# Symbolic Density Estimators for Unnormalized Distributions

## Abstract

Estimating the symbolic or analytical form of probability density functions (PDFs) from observed samples is a fundamental challenge in statistical and computational modelling. This process is critical for deriving interpretable and generalizable relationships characterizing the underlying phenomenon. Traditionally, this estimation depends strongly on domain expertise and prior field-specific knowledge, with experts selecting appropriate functional forms or parametric families based on empirical evidence and theoretical understanding. The coefficients of these forms are then typically determined through parameter estimation. In this paper, we develop a framework to estimate symbolic expressions of unnormalized distributions from their observed samples. We integrate deep generative models with symbolic regression (SR), incorporating inductive biases, such as factorizing large distributions, to keep the problem tractable. The deep generative models we examine include likelihood-based models, viz., flow models, and score-based models. Experiments show the effectiveness of the proposed framework for estimating density functions for multivariate toy distributions as well as lattices from computational physics, namely, XY model and $\phi^4$ theory. When applied to the renormalization problem in $\phi^4$ theory, the proposed framework identifies and recovers compact symbolic approximations of the action function directly from samples, yielding expressions that may be challenging to derive using traditional perturbative or analytic approaches in nonperturbative settings.

## 1 Introduction

In the physical sciences, understanding complex phenomena fundamentally depends on modelling the underlying system dynamics. The probability distributions that characterize these dynamics, as Hamiltonian or energy in a Boltzmann distribution, are traditionally formulated by domain experts, often in an analytical form with a set of parameters. These parameters can be estimated from the data using regression or data-fitting techniques (Shanahan et al., 2018). While effective, this process is labor-intensive and is heavily reliant on expert intuition. Consequently, the problem of automatically inferring an analytical or symbolic representation of a probability density function directly from observed data presents a significant challenge (Tohme et al., 2024). Addressing this problem lies at the intersection of machine learning and physics, offering the potential to automate model discovery and to enhance our understanding of complex physical systems. This problem is analogous to the system identification problem, widely studied in mechanical and control systems Sahoo et al. (2018), where the system dynamics is modeled in differential form (using differential equations) instead of an integral form (using Hamiltonian formulation).

Boltzmann distributions are abundantly used in physical sciences (Akhound-Sadegh et al., 2024; Midgley et al., 2023; Hasenfratz & Niedermayer, 1994). They have the form $p(\mathbf{x}) \propto \exp(-H(\mathbf{x}))$, where $H(\mathbf{x})$ is a scalar denoting energy of the system in state $\mathbf{x}$. Popular deep generative approaches to model these distributions include likelihood-based approaches, such as normalizing flows (Dinh et al., 2017) and autoregressive models (Uria et al., 2016; Van Den Oord et al., 2016), and energy-based approaches, such as score matching (Hyvärinen, 2005; Song & Ermon, 2019). These methods are able to generate samples from the target distribution while also providing information about the model likelihood i.e., log likelihood in case of likelihood based approaches and the gradient thereof in case of score matching approaches. However, these models are black box estimators. On the other hand, Symbolic regression (SR) methods are used to

estimate the equations of the input-output relations in a supervised way. Popular approaches for SR include genetic algorithms (Schmidt & Lipson, 2009; La Cava et al., 2016; Pal & Wang, 2017; La Cava et al., 2019; Cranmer, 2023), symbolic neural networks (Martius & Lampert, 2017; Sahoo et al., 2018; Kim et al., 2021) and large pre-trained networks (Biggio et al., 2021; Kamienny et al., 2022; Vastl et al., 2024), which work well for a small number of variables (typically up to 10). There are a few approaches (Cranmer et al., 2020) to combine SR with deep learning methods, which scale well with the number of variables, to simplify the SR task. While a few approaches (Tohme et al., 2024), attempt to recover expressions for low-dimensional unnormalized distributions, they rely on assuming the distribution's form.

In this paper, we propose a framework to integrate deep generative modelling with SR to estimate the density functions as symbolic expressions. We consider flow-based and score-based models for density estimation, and subsequently use SR methods, including genetic algorithms (Cranmer, 2023) and EQL-based approaches (Kim et al., 2021), to estimate the symbolic density function while incorporating task-specific inductive biases. The effectiveness of SR methods deteriorates notably with an increase in the number of variables or the dimensionality of the data, often leading to inaccurately estimated expressions (Cranmer et al., 2020; Biggio et al., 2021). For this, we employ a density factorization approach that leverages local dependencies within the distribution. This strategy effectively reduces the number of variables, thereby enabling more efficient and accurate expression estimation. However, deep generative models sample from high density regions of $p(x)$. For effective SR, which needs samples from both high and low density regions, we introduce noisy perturbation of samples.

We apply this framework to general density functions, such as multi-variate Gaussian and many-well distributions, as well as to problems in computational Physics, such as XY model (Kosterlitz & Thouless, 1973) and scalar $\phi^4$-theory (Singha et al., 2023; Albergo et al., 2019), and recover the energy functions, $H(\mathbf{x})$. We also derive the Hamiltonian for the renormalization of strongly-coupled $\phi^4$-theory, which is challenging to derive analytically. These experiments support the utility of the proposed framework for non-perturbative Physics, where analytic derivation of Hamiltonians is not feasible manually.

We emphasize that the primary contribution of this work is not the development of a new symbolic regression (SR) algorithm. Instead, we propose a general framework that integrates deep generative modeling with symbolic regression to recover interpretable energy functions and Hamiltonians from samples of unnormalized distributions. The central idea is to convert an unsupervised density estimation problem into a supervised symbolic recovery problem by leveraging quantities obtained from learned flow-based and score-based models, such as log-likelihoods and score fields. Standard SR methods can then be directly applied as modular components to obtain analytical approximations of the underlying energy landscape.

Within this framework, we introduce three key components that improve symbolic recoverability in practice. First, we employ a local factorization of the Hamiltonian, exploiting spatial locality in many-body systems to reduce the effective dimensionality of the symbolic search space and enable scalable recovery in high-dimensional settings. Second, we incorporate task-specific inductive biases through the choice of SR primitives and structural constraints, which encode prior knowledge about the underlying physical system. Third, we introduce a multi-scale perturbation strategy that augments samples from high-density regions produced by generative models with controlled noise, thereby improving coverage of low-density regions that are critical for stable symbolic recovery.

Overall, the proposed framework is complementary to existing advances in symbolic regression and symbolic distillation, and is designed to be compatible with a wide range of SR backends, including genetic-programming-based methods such as PySR and neural symbolic approaches such as EQL.

## 2 The Framework

Our framework consists of the following parts. (1) Training a deep generative model using the given samples. (2) Fitting an SR model using the given samples and the corresponding likelihood information (log values or gradients) provided by the deep generative model, and with density factorization conforming to the known nature of the system. The symbolic expression thus derived gives the energy $H(\mathbf{x})$ of the system. A block schematic of the framework is depicted in Fig. 1.

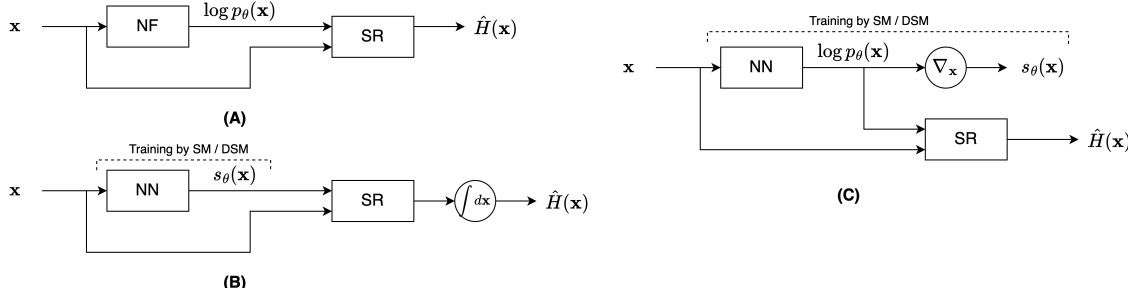

Figure 1: Block diagram of the proposed framework. (A) Flow-based estimation. (B) Score-based estimation. (C) Integration-free score-based estimation.

**Lattice Systems:** Apart from general distributions, such as multivariate Gaussian and many-well distributions, this paper focuses on lattice systems common in statistical and particle physics. Lattices are obtained by discretizing the continuous space on a regular grid. Recent works (Albergo et al., 2019; Singha et al., 2023; Kanaujia & Arora, 2025) predominantly use normalizing flows to model the lattice distributions because of the availability of exact likelihood from the flow model. In addition, Wang et al. (2023) have investigated diffusion-based approaches for the same purpose. In this work, we start with flow based models, and subsequently, introduce score-based methods to model the lattice systems.

**Problem statement:** Given the samples $\mathcal{D} = \{\mathbf{x}_i \in \mathcal{X}\}_{i=1}^N$ from an unknown distribution $p(\mathbf{x}); \mathbf{x} \in \mathbb{R}^d$, estimate a symbolic expression $\hat{H}(\mathbf{x}) \in \mathcal{H}$ such that $q_\theta(\mathbf{x}) = \frac{1}{Z} e^{-\hat{H}(\mathbf{x})}$ approximates $p(\mathbf{x})$.

**Symbolic Regression:** SR is a supervised learning approach that aims to discover an explicit mathematical expression describing the relationship between input and output variables. Given a dataset $\mathcal{D} = \{(\mathbf{x}_i, y_i)\}_{i=1}^N$, where $\mathbf{x}_i \in \mathbb{R}^n$ denotes the input variables, $y_i \in \mathbb{R}$ denotes the corresponding output, and $N$ is the number of observations, SR seeks to identify a function $f : \mathbb{R}^n \to \mathbb{R}$ such that

$$\hat{y}_i = f(\mathbf{x}_i), \tag{1}$$

where the functional form of $f$ is a concise closed-form mathematical expression.

Unlike conventional regression methods, which estimate the parameters of a predefined model, symbolic regression simultaneously searches for both the structure of the equation and its parameters (Schmidt & Lipson, 2009; 2010; Biggio et al., 2021). The search is typically performed over a predefined library of primitive operators and basis functions (e.g., $+$, $-$, $\times$, $\div$, sin, cos, polynomials), from which candidate expressions are constructed.

The objective of SR is to find an expression that accurately models the observed data while remaining interpretable. Let $\mathcal{F}$ denote the space of candidate symbolic expressions generated from the chosen primitive library. Given a loss function $\mathcal{L}$ that quantifies the discrepancy between the observed outputs and model predictions, SR seeks the optimal expression

$$f^* = \arg\min_{f \in \mathcal{F}} \mathcal{L}(f). \tag{2}$$

A common choice of loss function ($\mathcal{L}$) is the mean squared error (MSE),

$$\mathcal{L}_{\mathrm{MSE}} = \frac{1}{N} \sum_{i=1}^N (y_i - f(\mathbf{x}_i))^2. \tag{3}$$

In practice, additional criteria such as expression complexity, sparsity, or computational cost may also be incorporated into the objective to favor simpler and more interpretable expressions. The resulting symbolic

model provides both predictive accuracy and an explicit analytical form, making SR particularly attractive for scientific discovery and equation identification tasks (Dong & Zhong, 2025).

We use two approaches for SR, one is genetic programming based algorithm implemented using PySR library (Cranmer, 2023) and the other is deep neural network based equation learner (EQL) approach (Martius & Lampert, 2017; Sahoo et al., 2018; Kim et al., 2021). PySR uses evolutionary algorithms to optimise a tree data structure that represents a mathematical expression in order to best fit the data (Koza, 1994) in terms of mean squared error. In the search for the underlying structure, model error is usually balanced with model complexity to ensure brevity of the learned equation and avoids overfitting. PySR uses multi-population evolutionary genetic algorithm with multiple evolutions performed asynchronously. The other approach, EQL uses a symbolic neural network trained end-to-end through backpropagation. The model minimizes the mean squared error between $y$ and $\hat{y}$. To enforce sparsity, a smoothed variant of $L_{0.5}$ regularizer is used. Additional details for both SR approaches are provided in Appendix B.

The quality of convergence is indicated by the value of mean squared error. Many times, SR fails to converge and this may be indicated by a high value of mean squared error. One may use this to select the preferred solution from multiple SR models.

**Flow models:**  Flows model a target density $p(\mathbf{x})$ by applying a sequence of simple and learnable bijective transformations on samples from a known standard distribution $q(\mathbf{z}); \mathbf{z} \in \mathbb{R}^d$ (Papamakarios et al., 2021; Kobyzev et al., 2021; Rezende & Mohamed, 2015). In general, $q(\mathbf{z})$ is chosen to be standard Gaussian. Mathematically, a sample $\mathbf{z} \sim q(\mathbf{z})$ is transformed to $\mathbf{x}$ with a bijective map $T_\theta$, parameterized by $\theta$. The density of $\mathbf{x} = T_\theta(\mathbf{z})$ is given by

$$p_\theta(\mathbf{x}) = q(T_\theta^{-1}(\mathbf{x}))|\det(J_{T_\theta^{-1}}(\mathbf{x}))| \tag{4}$$

Here, $J_{T_\theta^{-1}}(\mathbf{x}) = \partial T_\theta^{-1}(\mathbf{x})/\partial \mathbf{x}$ is the Jacobian of $T_\theta^{-1}$. In order to model the unknown target distribution $p(\mathbf{x})$, using the samples from $p(\mathbf{x})$, $\theta$ is learnt by minimizing the KL divergence between the target density $p(\mathbf{x})$ and the model density $p_\theta(\mathbf{x})$ using gradient descent.

Once trained, the flow model can be used to obtain $\hat{y} = -\log p_\theta(\mathbf{x})$ for any $\mathbf{x}$, to form $\{(\mathbf{x}_i, \hat{y}_i)\}$ pairs to train the SR models, which converts the unsupervised density estimation problem to a supervised SR problem. An SR model estimates an expression $\hat{y} = f(\mathbf{x})$, which is related to the energy function as

$$f(\mathbf{x}) = \hat{H}(\mathbf{x}) + \log \hat{Z} \tag{5}$$

The expression for $\hat{H}(\mathbf{x})$ is obtained by dropping the terms in $f(\mathbf{x})$ which are independent of $\mathbf{x}$.

**Score Matching:**  Score function for any distribution $p(\mathbf{x})$ is defined as $s(\mathbf{x}) = \nabla_\mathbf{x} \log p(\mathbf{x})$. An advantage of using $s(\mathbf{x})$ (over $\log p(\mathbf{x})$) is that it is independent of the normalizing constant $Z$. In score matching (Song & Ermon, 2019; Song et al., 2020), a deep model $s_\theta(\mathbf{x})$, parameterized by $\theta$, estimates the target score $s(\mathbf{x})$. Training is carried out by minimizing the expected square distance,

$$J(\theta) = \frac{1}{2}\mathbb{E}_{p(\mathbf{x})}\|s_\theta(\mathbf{x}) - s(\mathbf{x})\|^2 \tag{6}$$

In the absence of $s(\mathbf{x})$, $J$ can be simplified under certain regularity conditions as detailed in Hyvärinen (2005) by

$$J(\theta) = \mathbb{E}_{p(\mathbf{x})}[\text{Tr}(\nabla_x s_\theta(\mathbf{x})) + \frac{1}{2}\|s_\theta(\mathbf{x})\|^2] \tag{7}$$

where Tr is the trace. Once trained, the model can estimate the score $s_\theta(\mathbf{x}) = \nabla_\mathbf{x} \log p_\theta(\mathbf{x})$ for any $\mathbf{x}$ to form pairs $\{(\mathbf{x}_i, \hat{y}_i = -s_\theta(\mathbf{x}_i))\}$ to train the SR models.

The SR model estimates the expression $\hat{y} = f(\mathbf{x})$, which is related to the energy function as

$$f(\mathbf{x}) = \nabla_\mathbf{x}\hat{H}(\mathbf{x}) \tag{8}$$

which is free from the normalizing constant. In principle, recovering $\hat{H}(\mathbf{x})$ from $f(\mathbf{x})$ requires the learned vector field to be conservative (i.e., curl-free), such that it can be expressed as the gradient of a scalar

potential. In our setting, the score model is trained to approximate the gradient of the log-density, which is theoretically conservative for the target distributions considered, although approximation errors may lead to small deviations in practice. Finally, $\hat{H}(\mathbf{x})$ is estimated by integrating $f(\mathbf{x})$ with respect to $\mathbf{x}$.

**Denoising Score Matching:** Score matching does not scale well to high dimensional $\mathbf{x}$ due to the high cost involved in computing $\text{Tr}(\nabla_x s_\theta(\mathbf{x}))$. To address this, Vincent (2011) proposed denoising score matching approach that perturbs $\mathbf{x}$ with a pre-defined noise distribution $q_\sigma(\tilde{\mathbf{x}}|\mathbf{x})$, where, $\tilde{\mathbf{x}} = \mathbf{x} + \epsilon$ and $\epsilon \sim q_\sigma(\tilde{\mathbf{x}}|\mathbf{x}) = \mathcal{N}(\tilde{\mathbf{x}}|\mathbf{x}, \sigma^2 I)$. The score of the perturbed data distribution $q_\sigma(\tilde{\mathbf{x}})$ is learned by minimizing

$$J(\theta; \sigma) = \frac{1}{2}\mathbb{E}_{q_\sigma(\mathbf{x}, \tilde{\mathbf{x}})}[\|s_\theta(\tilde{\mathbf{x}}) - \nabla_{\tilde{\mathbf{x}}} \log q_\sigma(\tilde{\mathbf{x}}|\mathbf{x})\|_2^2] \tag{9}$$

For small $\sigma$, we get $s_\theta(\mathbf{x}) \approx \nabla_{\mathbf{x}} \log p(\mathbf{x})$. Training is carried out across multiple noise levels, which generates samples from low-density regions of the $p(\mathbf{x})$, thereby improving score estimation.

Once trained, the score model $s_\theta(\mathbf{x})$ can be used to estimate $\hat{H}(\mathbf{x})$ in the same way as described above in the score matching approach.

**Mitigating Integration Issues:** In score-based models, SR estimates the score function as a symbolic expression, which is subsequently integrated to estimate $\hat{H}(\mathbf{x})$ (see Eq. (8)). However, when SR yields a complex expression for score, its integration becomes analytically challenging. To ameliorate this issue, we propose a reformulation of the modelling approach: instead of modelling the score function, we model $H_\theta(\mathbf{x})$ using a neural network. The score is then obtained as the negative gradient obtained by autograd, $s_\theta(\mathbf{x}) = -\nabla_{\mathbf{x}} H_\theta(\mathbf{x})$. This formulation enables training via standard objectives such as score matching or denoising score matching, as defined in Eqs. (7) and (9).

**Factorization:** The effectiveness of SR methods reduces drastically with the increase in the size of search space (the number of variables and operations). The lattice models we apply our framework to are large. However, most physical lattices have local interactions which can be leveraged to simplify the SR problem. There are existing approaches leveraging this property for Monte Carlo sampling of lattices (Kennedy & Pendleton, 1985; Faraz et al., 2024). We use the same property for SR, however, allowing the local interactions to extend to any number of neighbours, makes our approach more generalizable.

Given a lattice $\mathbf{x}$ (a matrix, say), the true energy function can be factorized as

$$H(\mathbf{x}) = \sum_l h(x_l|x_{n(l)}) \tag{10}$$

where, $h$ is the local energy function, $x_l$ is the value of the field on lattice site $l$, and $n(l)$ is the set of neighbors of site $l$. Fig. 2(a) illustrates this.

During training with SR, $\mathbf{x}$ is factorized into patches and $\hat{H}$ is estimated as

$$\hat{H}(\mathbf{x}) = \sum_l \hat{h}(x_{p(l)}) \tag{11}$$

where $p(l)$ is the set of lattice sites in the patch anchored at site $l$. Overall, SR estimates $\hat{h}$ over variables $x_{p(l)}$ which are small in number. Fig. 2(b) shows patches of different sizes.

It is to be noted that there is redundancy in this representation. Multiple representations of $\hat{h}(x_{p_l})$ may lead to the same $\hat{H}(\mathbf{x})$. For instance, with a $3 \times 3$ patch shown in Fig. 2(c), any $\hat{h}(x_{p_l}) = \sum_{i=0}^8 a_i g(x_i)$, for any value of $\{a_i\}$ with $\sum_{i=0}^8 a_i = a$ being constant, leads to the same $\hat{H}(\mathbf{x}) = \sum_l a g(x_l)$. Here, $g$ could be any univariate function. In general, $g$ could be multivariate; e.g., $\hat{h}'(x_{p_l}) = g(x_0, x_4, x_6)$ and $\hat{h}''(x_{p_l}) = g(x_1, x_5, x_7)$, with the same relative positions of lattice sites, lead to the same $\hat{H}(\mathbf{x})$.

In Fig. 1, methods (A) and (C), SR minimizes $\mathbb{E}_{\mathbf{x}}[\|\hat{y}(\mathbf{x}) - \sum_l \hat{h}(x_{p(l)})\|^2]$ to estimate $\hat{h}$, where $\hat{y}(\mathbf{x}) = -\log p_\theta(\mathbf{x})$ is the target estimated by the deep model. It has no target for individual $\hat{h}(x_{p(l)})$ but only matches the sum over the entire lattice. On the other hand, in method (B), SR minimizes $\mathbb{E}_{\mathbf{x}, l}[\|s_\theta(\mathbf{x})[l] - f(x_{p(l)})\|^2]$ to estimate $f$ (Eq. equation 8) with target as the $l^{th}$ element of matrix $s_\theta(\mathbf{x})$ estimated by the deep model.

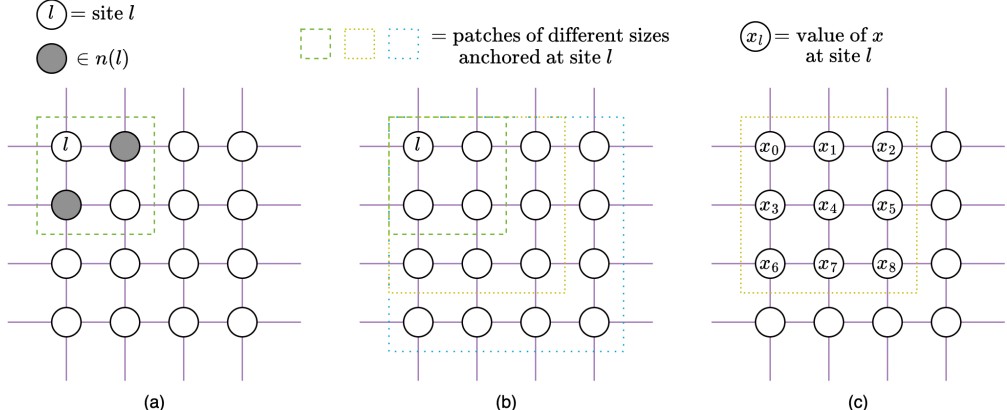

Figure 2: Lattice Factorization. (a) True $H(\mathbf{x})$ can be factorized with a $2 \times 2$ patch. (b) While inference with SR, different patch sizes provide different inductive biases. (c) Illustration for redundancy in representation.

Note that there is no redundancy of representation here as we are modelling site-wise gradients and not the lattice likelihood.

**Learning from low density regions:** Robust SR needs samples from both high and low values of $H(\mathbf{x})$. The original samples $\mathbf{x} \sim p(\mathbf{x})$ represent regions with low energy, but high energy regions are conspicuous by their absence. Hence, we append the original set $\mathcal{D}$ with noisy perturbations $\mathbf{x}' = \mathbf{x} + \epsilon$ where $\epsilon \sim \mathcal{N}(\mathbf{0}; \sigma^2 I)$. Then, we estimate $\log p_\theta(\mathbf{x}')$ or $s_\theta(\mathbf{x}')$ using the deep models, and thus, form the input-output pairs needed for SR.

Algorithm 1 outlines the training procedure for symbolic estimation of unnormalized distributions, combining deep generative modelling with SR. The method first learns a density or score model from data, augments low-density regions, and subsequently recovers an analytical expression for the Hamiltonian via SR.

## 3 Case Studies

In this section we describe five specific examples where we apply our framework.

**Multivariate Gaussian density.**

$$p(\mathbf{x}; \mu, \mathbf{\Sigma}) = \frac{1}{(2\pi)^{d/2} |\mathbf{\Sigma}|^{1/2}} e^{-\frac{1}{2} (\mathbf{x} - \mu)^T \mathbf{\Sigma}^{-1} (\mathbf{x} - \mu)} \tag{12}$$

Fixing $\mu = [-1.5, 1.5]^T$, we use two variants of $\mathbf{\Sigma}$ with $\mathbf{x} \in \mathbb{R}^2$. In the first variant, we use a diagonal $\mathbf{\Sigma}$ as $I$, resulting in $H(\mathbf{x}) = 0.5(x_1 + 1.5)^2 + 0.5(x_2 - 1.5)^2$. In the second one, we use a full $\mathbf{\Sigma}$ as $\begin{bmatrix} 1.0 & 0.5 \\ 0.5 & 1.0 \end{bmatrix}$, leading to $H(\mathbf{x}) = 0.5(x_1 + 1.5)^2 + 0.5(x_2 - 1.5)^2 + 0.5(x_1 + 1.5)(x_2 - 1.5)$. We apply our framework to the data sampled from these distributions, and expect it to recover $H(\mathbf{x})$.

**Many Well Distribution ($d$-dimensional).** It is a synthetic distribution given by the product of $d/2$ copies of the 2-D double well distribution (Noé et al., 2019; Wu et al., 2020). The Hamiltonian of 2-D double well distribution is

$$H(\mathbf{x}) = x_1^4 - 6x_1^2 - 0.5x_1 + 0.5x_2^2, \qquad \mathbf{x} \in \mathbb{R}^2 \tag{13}$$

and that of $d$-dimensional many well distribution is

$$H(\mathbf{x}) = \sum_{i=1}^{d/2} \left( x_{2i-1}^4 - 6x_{2i-1}^2 - 0.5x_{2i-1} + 0.5x_{2i}^2 \right), \qquad \mathbf{x} \in \mathbb{R}^d \tag{14}$$

---

**Algorithm 1** Symbolic Function Estimation for Unnormalized Distributions

---

1: **Input:** Data $\mathcal{D} = \{\mathbf{x}_i\}_{i=1}^N$ from distribution
2: **Stage 1: Training the Generative Model**
3: Initialise: Parameters $\theta$ of estimator $M$ (NF–based or score-based)
4: **while** $\mathcal{L}_M(\theta)$ not converged **do**
5:     Sample a mini-batch from $\mathcal{D}$
6:     Compute loss $\mathcal{L}_M(\theta)$
7:     Update parameters:

$$\theta \leftarrow \theta - \eta \nabla_\theta \mathcal{L}_M(\theta)$$

8: **end while**
9: **Stage 2: Low-density Sample Augmentation**
10: Generate perturbed samples

$$\mathbf{x}' \leftarrow \mathbf{x} + \boldsymbol{\epsilon}, \quad \boldsymbol{\epsilon} \sim \mathcal{N}(\mathbf{0}, \sigma^2 I)$$

11: Construct augmented dataset $\mathcal{D}_{\text{aug}} = \mathcal{D} \cup \{\mathbf{x}'_i\}_{i=1}^N$
12: **Stage 3: Label Construction**
13: Evaluate the trained estimator $M$ to obtain predicted quantities

$$\hat{y}_i \in \{-\text{NLL}(\mathbf{x}_i), \ -\hat{s}(\mathbf{x}_i)\}$$

14: Form regression dataset

$$\mathcal{D}' = \{(\mathbf{x}_i, \hat{y}_i)\}$$

15: **Stage 4: Symbolic Regression**
16: Initialise: Parameters of the SR model
17: **if** local factorisable distribution **then**
18:     Get local patches from $\mathcal{D}'$          ▷ Use local Model Factorization
19: **end if**
20: Optimise SR parameters on $\mathcal{D}'$ using MSE loss
21: Recover analytical expression $\hat{y} = f(\mathbf{x})$
22: Estimate Hamiltonian $\hat{H}(\mathbf{x})$ from $f(\mathbf{x})$
23: **Return:** $\hat{H}(\mathbf{x})$

---

We conduct experiments for $d = 2, 4, 8, 16, 32, 64$, expecting the framework to recover $H(\mathbf{x})$.

**XY Model.** It is a theoretical model used in statistical mechanics (Kosterlitz & Thouless, 1973) to describe two-dimensional systems of interacting spins and to study phase transitions. It is widely employed to model Boltzmann distributions on lattices (Kanaujia et al., 2024; Singh et al., 2021; Beach et al., 2018). Here, $\mathbf{x} \in [0, 2\pi)^d$ represents the spin configuration on a square 2-D grid with $d$ sites, and the energy of the system is given by

$$H(\mathbf{x}) = -0.5\lambda \sum_{l=1}^d \sum_{l' \in n(l)} \cos(x_l - x_{l'}) \tag{15}$$

Here, $\lambda$ is the coupling constant that quantifies interaction strength and $n(l)$ is a set of two neighbors (right and below, cf. Fig. 2(a)) of site $l$. We consider a grid of size $8 \times 8$, i.e., $d = 64$, with $\lambda = 1/1.4$. A patch $p$ of size $2 \times 2$ is used to factorize $\hat{H}(\mathbf{x})$.

**Scalar $\phi^4$ theory.** It is a widely used in computational Physics (Albergo et al., 2019; Singha et al., 2023) to study scalar fields. With $\mathbf{x} \in \mathbb{R}^d$ as a scalar field on a square 2-D lattice with $d$ sites, the energy function (or action) is given by

$$H(\mathbf{x}) = \sum_{l=1}^d \left( \lambda_1 x_l^4 + \lambda_2 x_l^2 + 2 \sum_{l' \in n(l)} (x_l^2 - x_l x_{l'}) \right) \tag{16}$$

where $n(l)$ denotes the two neighboring sites (right and below; see Fig. 2(a)). Here, $\lambda_1$ is the coupling constant and $\lambda_2$ represents the squared mass of the scalar particle associated with the field. We conduct experiments on lattices of size $8 \times 8$, i.e., $d = 64$, at two different values of $\lambda_2$, namely $-4$ and $1$, while $\lambda_1$ is fixed at 4. Here, a $2 \times 2$ patch $p$ is sufficient to recover $H(\mathbf{x})$ but we also experiment with patches of different sizes. Larger patches allow recovering distant interactions but increase the number of variables, making it difficult for SR to recover the true equation.

**Renormalization.** Renormalization in statistical Physics is akin to zooming out in image processing. It enables one to study the phenomena on larger scales by reducing the resolution of lattices (called coarse-graining) for computational efficiency. Empirically, a finer lattice is coarsened by some kind of pooling; generally average pooling is used. A fine lattice of size $\sqrt{d} \times \sqrt{d}$, on pooling with a $2 \times 2$ filter, reduces to a coarser lattice of size $\sqrt{d}/2 \times \sqrt{d}/2$.

Traditionally, the effective action $H_{\text{eff}}(\mathbf{x})$ at the coarser scale is estimated analytically by integrating out terms in the original action $H(\mathbf{x})$. The effectiveness of this method is substantially limited by the extent of available analytical techniques. Our framework, however, allows one to estimate $H_{\text{eff}}(\mathbf{x})$ directly from coarse-grained lattice samples.

We test our framework to find $H_{\text{eff}}(\mathbf{x})$ for highly correlated $\phi^4$ theory with $\lambda_1 = 4$ and $\lambda_2 = 1$, which lies in the non-perturbative regime where perturbative analysis fails. We start with $64 \times 64$ lattice samples from Eq. (16), i.e., $d = 4096$, and coarse-grain them to $32 \times 32$ lattice samples by average pooling. These coarse-grain samples are then used as $\mathcal{D}$ to estimate $H_{\text{eff}}(\mathbf{x})$. These samples are progressively coarse-grained to $16 \times 16$, and then, to $8 \times 8$ to see how $H_{\text{eff}}(\mathbf{x})$ evolves to those lattice scales.

## 4 Experiments and Results

Most existing methods for symbolic density estimation assume a relatively simple parametric form for the underlying probability density function. To the best of our knowledge, MeSSY (Tohme et al., 2024) is one of the few existing methods specifically designed for symbolic density estimation that is applicable to our setting, and we therefore use it as the primary baseline. MeSSY constructs an ansatz for the energy of the target distribution as a weighted linear combination of predefined basis functions, i.e., a set of linearly independent functions (e.g., $x$, $x^2$, $\sin(x)$, $\cos(x)$). The corresponding weights are estimated by maximizing the log-likelihood over the available samples. While we also investigated scaling MeSSY to higher-dimensional problems, its reliance on numerical integration and rapidly growing polynomial basis functions resulted in prohibitive computational costs, limiting its applicability to very low-dimensional settings in our experiments. Consequently, MeSSY is reported only for cases where computation remained tractable.

In contrast, the proposed approach integrates deep generative models with symbolic regression to estimate the analytic expression of density, using a rich set of operations (e.g., $\times, \sin, \cos, \log, \exp$) and relatively simpler basis functions. This enables modelling complicated energy functions (e.g., XY model), and is not limited to only linear combinations of basis functions.

In this section, we present experimental results across a range of datasets, spanning from low-dimensional distributions ($d = 2$) to high-dimensional ones ($d = 1024$). These experiments validate the effectiveness of the proposed framework, highlighting its ability to handle increasing complexity and scale while maintaining performance. We employ normalizing flows and score-based models as the underlying generative components, which are coupled with two distinct SR frameworks, viz., PySR (genetic algorithm driven) (Cranmer, 2023) and EQL (neural network based) (Kim et al., 2021), to form the integrated pipeline.

The models are evaluated from two complementary perspectives: (i) predictive accuracy of the recovered Hamiltonian and (ii) fidelity of the recovered symbolic expression. To assess predictive accuracy, we compare the estimated Hamiltonian $\hat{H}(\mathbf{x})$ with the ground-truth Hamiltonian $H(\mathbf{x})$ using the mean squared error (MSE) and the coefficient of determination ($R^2$). The $R^2$ score is defined as

$$R^2(H, \hat{H}; \mathcal{D}) = 1 - \frac{\sum_{\mathbf{x} \in \mathcal{D}} \left( H(\mathbf{x}) - \hat{H}(\mathbf{x}) \right)^2}{\sum_{\mathbf{x} \in \mathcal{D}} \left( H(\mathbf{x}) - \bar{H} \right)^2}, \tag{17}$$

where $\bar{H} = \mathbb{E}_{\mathbf{x} \sim \mathcal{D}}[H(\mathbf{x})]$ denotes the empirical mean of the true Hamiltonian over the dataset $\mathcal{D}$. Lower MSE and higher $R^2$ values indicate closer agreement between $\hat{H}(\mathbf{x})$ and $H(\mathbf{x})$. We additionally present the recovered symbolic expressions for qualitative comparison.

While these metrics quantify predictive performance, the primary objective of the proposed framework is to recover an interpretable symbolic representation of the underlying Hamiltonian. Consequently, MSE and $R^2$ alone do not fully characterize symbolic recovery quality, as distinct symbolic expressions may achieve similar predictive accuracy despite differing in structure, coefficient values, or the presence of spurious interactions. To directly evaluate symbolic recovery, we report the following additional metrics.

**Relative Coefficient Error (RCE).** For symbolic terms that are correctly recovered, we measure coefficient accuracy using the average relative coefficient error. Let $\mathcal{M}$ denote the set of matched symbolic terms, $c_i$ the ground-truth coefficient, and $\hat{c}_i$ the recovered coefficient. The RCE is defined as

$$\text{RCE} = \frac{1}{|\mathcal{M}|} \sum_{i \in \mathcal{M}} \frac{|\hat{c}_i - c_i|}{|c_i|}. \tag{18}$$

An RCE of zero corresponds to exact coefficient recovery.

**Precision (P).** Precision measures the fraction of recovered symbolic terms that correspond to true terms in the target Hamiltonian,

$$\text{Precision} = \frac{\text{TP}}{\text{TP} + \text{FP}}, \tag{19}$$

where TP denotes the number of correctly recovered terms (true positives) and FP denotes the number of spurious terms (false positives).

**Recall (R).** Recall measures the fraction of ground-truth symbolic terms that are successfully recovered,

$$\text{Recall} = \frac{\text{TP}}{\text{TP} + \text{FN}}, \tag{20}$$

where FN denotes the number of ground-truth terms that are not recovered.

**F1 Score.** The F1 score summarizes precision and recall through their harmonic mean,

$$\text{F1} = \frac{2PR}{P + R}. \tag{21}$$

An F1 score of one indicates perfect symbolic recovery.

**Number of Spurious Terms (ST).** We also report the number of symbolic terms present in the recovered expression that do not appear in the ground-truth Hamiltonian. This metric provides a direct measure of symbolic overfitting and highlights cases where good predictive performance is achieved through physically irrelevant interactions.

Together, these metrics provide a comprehensive assessment of symbolic recovery quality by quantifying predictive accuracy, coefficient estimation accuracy, structural correctness, and the presence of spurious interactions.

**Multivariate Gaussian density:** We train both flow-based and score-matching models on $2.5 \times 10^4$ samples drawn from a two-dimensional Gaussian distribution. For symbolic regression (SR) with the EQL model, all $2.5 \times 10^4$ samples are used for training. The proposed approach successfully recovers the true functional form of $H(\mathbf{x})$. Evaluation on an independent test set comprising $2 \times 10^4$ samples yields a mean

Table 1: Results and estimated expressions $\hat{h}([x_1, x_2]; \lambda)$ for a 2-D Gaussian distribution using various methods. The mean vector is fixed at $\mu = [-1.5, 1.5]^T$, with two different covariance matrices, $\Sigma_1$ and $\Sigma_2$, as detailed in the table. The target functions are: (A) $0.5x_1^2 + 0.5x_2^2 + 1.5x_1 - 1.5x_2$, and (B) $0.67x_1^2 - 0.67x_1x_2 + 0.67x_2^2 + 3.0x_1 - 3.0x_2$. Incorrect terms are shown in gray. NF = Flow model, SM = Score matching.

| | | MSE($\downarrow$) | $R^2$($\uparrow$) | RCE($\downarrow$) | ST($\downarrow$) | P($\uparrow$) | R($\uparrow$) | F1($\uparrow$) | Estimated $\hat{h}(x_{p(l)})$ |
|---|---|---|---|---|---|---|---|---|---|
| $\Sigma = \begin{bmatrix} 1.0 & 0.0 \\ 0.0 & 1.0 \end{bmatrix}$ | MeSSY | 0.0114 | 0.99 | 0.0433 | 1 | 0.80 | 1.00 | 0.89 | $0.48x_1^2 - \color{gray}{0.02x_1x_2} + 0.50x_2^2 + 1.39x_1 - 1.41x_2$ |
| | NF + PySR | 0.0001 | 0.99 | 0.0017 | 0 | 1.00 | 1.00 | 1.00 | $0.5x_1^2 + 0.5x_2^2 + 1.5x_1 - 1.49x_2$ |
| | NF + EQL | 0.0011 | 0.99 | 0.0033 | 0 | 1.00 | 1.00 | 1.00 | $0.5x_1^2 + 0.5x_2^2 + 1.52x_1 - 1.5x_2$ |
| | SM + PySR | 0.0013 | 0.99 | 0.0150 | 0 | 1.00 | 1.00 | 1.00 | $0.51x_1^2 + 0.49x_2^2 + 1.53x_1 - 1.5x_2$ |
| | SM + EQL | 0.0014 | 0.99 | 0.0230 | 0 | 1.00 | 1.00 | 1.00 | $0.49x_1^2 + 1.47x_1 + 0.52x_2^2 - 1.52x_2$ |
| $\Sigma = \begin{bmatrix} 1.0 & 0.5 \\ 0.5 & 1.0 \end{bmatrix}$ | MeSSY | 0.0493 | 0.95 | 0.1199 | 0 | 1.00 | 1.00 | 1.00 | $0.72x_1^2 - 0.85x_1x_2 + 0.63x_2^2 + 3.38x_1 - 3.21x_2$ |
| | NF + PySR | 0.0006 | 0.99 | 0.0057 | 0 | 1.00 | 1.00 | 1.00 | $0.67x_1^2 - 0.66x_1x_2 + 0.67x_2^2 + 2.98x_1 - 2.98x_2$ |
| | NF + EQL | 0.0033 | 0.99 | 0.0093 | 0 | 1.00 | 1.00 | 1.00 | $0.67x_1^2 - 0.68x_1x_2 + 0.66x_2^2 + 3.02x_1 - 2.97x_2$ |
| | SM + PySR | 0.0047 | 0.99 | 0.0109 | 0 | 1.00 | 1.00 | 1.00 | $0.67x_1^2 - 0.70x_1x_2 + 0.67x_2^2 + 2.99x_1 - 3.02x_2$ |
| | SM + EQL | 0.0004 | 0.99 | 0.0093 | 0 | 1.00 | 1.00 | 1.00 | $0.66x_1^2 - 0.67x_1x_2 + 0.66x_2^2 + 2.97x_1 - 2.98x_2$ |

squared error (MSE) close to zero and a coefficient of determination ($R^2$) close to one. In comparison, PySR attains comparable performance using only $10^3$ samples. These results hold consistently for both Gaussian distributions considered—those with diagonal and non-diagonal covariance matrices $\mathbf{\Sigma}$. Both methods recover the true $H(\mathbf{x})$ without the need for noise perturbation to account for low-density regions. The quantitative results are presented in Table 1.

Experiments with the baseline, MeSSY, also yield MSE values close to zero and $R^2$ near one; however, the recovered expression deviates from the ground truth. In particular, the coefficients of several polynomial terms differ from those in the true $H(\mathbf{x})$, and the predicted expression contains additional spurious multinomial terms.

**Many Well Distribution ($d$-dimensional):** We train flow, score-matching, and denoising score-matching models with $10^5$ samples from MW distribution. For SR, we employ factorisation using patches of size $2 \times 1$ to reduce the search space with this inductive bias. Evaluation is performed on $2 \times 10^4$ test samples, and results are summarised in Table 2.

For lower dimensions ($d = 2, 4, 8$), both flow and score-matching models, as well as SR approaches (PySR and EQL), successfully recover the true $H(\mathbf{x})$. EQL is trained using all $10^5$ samples, whereas PySR achieves comparable performance with only $10^3$ samples. No noise perturbation is required.

For higher dimensions ($d = 16, 32, 64$), the denoising score-matching model outperforms the flow model, likely due to the highly multimodal nature of $p(\mathbf{x})$ with $2^{d/2}$ modes. Flow models trained with forward KL divergence are susceptible to mode-covering behavior (Midgley et al., 2023; Kanaujia et al., 2024). In these cases, PySR also surpasses EQL, reflecting the advantage of the broader search capability of genetic algorithms over the gradient-based optimization used in EQL.

While MeSSY is able to recover the general structure of $H(\mathbf{x})$ for $d = 2$, it fails to converge and recover an expression for $d > 2$. By "failure to converge," we mean that the optimization procedure of MeSSY does not terminate successfully within a reasonable computational budget. In our experiments, the algorithm repeatedly continued its search/optimization loop without producing a stable symbolic expression, even when trained using a very small number of samples for datasets with dimensions greater than two ($d > 2$). However, even for $d = 2$, the coefficients of several terms differ from those in the true $H(\mathbf{x})$, and the recovered expression includes additional multinomial terms (cf. Table 2).

**XY Model:** We experiment on lattices of size $8 \times 8$ ($d = 64$) with $\lambda = 1/1.4$. The approximate energy function $\hat{H}(\mathbf{x})$ is factorized using patches of size $2 \times 2$ (c.f. Fig. 2(a)). We use $10^5$ samples for training flow and score-based models. For score-based models, we find that parameterizing $H_\theta(\mathbf{x})$ with a neural network and obtaining $\nabla_{\mathbf{x}} H_\theta(\mathbf{x})$ via automatic differentiation yields better performance than directly modelling $s_\theta(\mathbf{x})$, as the latter approach suffers from integration issues due to nested trigonometric terms.

Denoising score matching outperforms standard score matching, while overall the flow model achieves superior results compared to score-based methods. Among symbolic regression approaches, PySR consistently

Table 2: Results for Many Well distribution with various dimensions, $d = 2, 4, 8, 16, 32, 64$. The true function is $h(x_{p_l}) = x_1^4 - 6x_1^2 - 0.5x_1 + 0.5x_2^2$. Estimated terms that do not match the true function are shown in gray. NF = Flow model, SM = Score matching.

| $d$ | Model | MSE($\downarrow$) | $R^2$($\uparrow$) | RCE($\downarrow$) | ST($\downarrow$) | P($\uparrow$) | R($\uparrow$) | F1($\uparrow$) | Estimated Equation $\hat{h}(x_{p(l)})$ |
|---|---|---|---|---|---|---|---|---|---|
| | MeSSY | 0.02 | 0.98 | 0.2396 | 2 | 0.67 | 1.00 | 0.80 | $0.96x_1^4 + 0.1x_1^3 - 5.77x_1^2 - 0.83x_1 + 0.61x_2^2 - 0.23x_2$ |
| | NF + PySR | 0.00 | 0.99 | 0.0008 | 0 | 1.00 | 1.00 | 1.00 | $x_1^4 - 6.02x_1^2 - 0.5x_1 + 0.5x_2^2$ |
| 2 | NF + EQL | 0.00 | 0.99 | 0.0196 | 0 | 1.00 | 1.00 | 1.00 | $0.99x_1^4 - 5.95x_1^2 - 0.49x_1 + 0.52x_2^2$ |
| | SM + PySR | 0.00 | 0.99 | 0.0142 | 0 | 1.00 | 1.00 | 1.00 | $0.98x_1^4 - 5.9x_1^2 - 0.51x_1 + 0.5x_2^2$ |
| | SM + EQL | 0.00 | 0.99 | 0.0617 | 0 | 1.00 | 1.00 | 1.00 | $0.99x_1^4 - 5.9x_1^2 - 0.59x_1 + 0.48x_2^2$ |
| | NF + PySR | 0.00 | 0.97 | 0.0296 | 0 | 1.00 | 1.00 | 1.00 | $1.02x_1^4 - 6.11x_1^2 - 0.52x_1 + 0.52x_2^2$ |
| 4 | NF + EQL | 0.00 | 0.99 | 0.0463 | 0 | 1.00 | 1.00 | 1.00 | $1.0x_1^4 - 6.03x_1^2 - 0.47x_1 + 0.46x_2^2$ |
| | SM + PySR | 0.01 | 0.89 | 0.0188 | 0 | 1.00 | 1.00 | 1.00 | $1.04x_1^4 - 6.21x_1^2 - 0.5x_1 + 0.5x_2^2$ |
| | SM + EQL | 0.48 | 0.75 | 0.0963 | 0 | 1.00 | 1.00 | 1.00 | $1.0x_1^4 - 6.03x_1^2 - 0.69x_1 + 0.50x_2^2$ |
| | NF + PySR | 0.00 | 0.90 | 0.0508 | 0 | 1.00 | 1.00 | 1.00 | $1.0x_1^4 - 6.02x_1^2 - 0.55x_1 + 0.55x_2^2$ |
| 8 | NF + EQL | 0.23 | 0.96 | 0.2900 | 4 | 0.43 | 0.75 | 0.75 | $0.92x_1^4 - 5.58x_1^2 - 0.86x_1 + 0.14x_1^2x_2^2 + 0.11x_1^2x_2 + 0.24x_1x_2 - 0.2x_2$ |
| | SM + PySR | 0.01 | 0.99 | 0.0213 | 0 | 1.00 | 1.00 | 1.00 | $x_1^4 - 6.03x_1^2 - 0.46x_1 + 0.5x_2^2$ |
| | SM + EQL | 0.42 | 0.93 | 0.0050 | 1 | 0.75 | 0.75 | 0.75 | $0.99x_1^4 - 0.11x_1^3 - 5.97x_1^2 + 0.5x_2^2$ |
| | NF + PySR | 0.51 | 0.96 | 0.1304 | 0 | 1.00 | 1.00 | 1.00 | $1.0x_1^4 - 6.01x_1^2 - 0.67x_1 + 0.59x_2^2$ |
| 16 | NF + EQL | 0.67 | 0.94 | 0.1908 | 2 | 0.67 | 1.00 | 0.80 | $0.78x_1^4 - 4.78x_1^2 - 0.53x_1 + 0.64x_2^2 + 0.11x_1^2x_2 - 0.26x_2$ |
| | SM + PySR | 0.65 | 0.94 | 0.1433 | 1 | 0.80 | 1.00 | 0.89 | $1.26x_1^4 - 5.80x_1^2 - 0.36x_1 + 0.50x_2^2 - 0.06x_1^6$ |
| | SM + EQL | 0.06 | 0.99 | 0.0471 | 0 | 1.00 | 1.00 | 1.00 | $x_1^4 - 5.95x_1^2 - 0.41x_1 + 0.5x_2^2$ |
| | NF + PySR | 4.63 | 0.80 | 0.3663 | 0 | 1.00 | 1.00 | 1.00 | $0.71x_1^4 - 4.47x_1^2 - 0.20x_1 + 0.34x_2^2$ |
| 32 | NF + EQL | 5.29 | 0.77 | 0.7539 | 5 | 0.38 | 0.75 | 0.50 | $0.35x_1^4 - 2.93x_1^2 + 1.05x_2^2 - 0.33x_1^3x_2 - 0.33x_1^2x_2^2 - 0.11x_1^2x_2 + 0.48x_1x_2 + 0.36x_2$ |
| | SM + PySR | 0.48 | 0.98 | 0.1304 | 1 | 0.80 | 1.00 | 0.89 | $0.79x_1^4 - 4.73x_1^2 - 0.45x_1 + 0.50x_2^2 - 0.02x_2$ |
| | SM + EQL | 0.69 | 0.99 | 0.3634 | 3 | 0.57 | 1.00 | 0.73 | $0.87x_1^4 - 5.63x_1^2 - 1.13x_1 + 0.49x_2^2 + 0.11x_1^6 - 0.18x_1^5 + 0.54x_1^3$ |
| | NF + PySR | 1.67 | 0.93 | 0.7467 | 3 | 0.57 | 1.00 | 0.73 | $0.13x_1^4 - 2.06x_1^2 + 1.16x_1 + 0.43x_2^2 + 0.04x_1^6 + 0.19x_1^5 - 1.14x_1^3$ |
| 64 | NF + EQL | 3.18 | 0.93 | 1.0379 | 2 | 0.67 | 1.00 | 0.80 | $-0.93x_1^4 + 0.13x_1^2 - 0.21x_1 + 0.19x_2^2 + 0.12x_1^3x_2 - 0.48x_1x_2$ |
| | SM + PySR | 0.53 | 0.99 | 0.1358 | 3 | 0.57 | 1.00 | 0.73 | $1.37x_1^4 - 6.56x_1^2 - 0.47x_1 + 0.51x_2^2 - 0.06x_1^6 - 0.02x_1^2x_2 + 0.05x_2$ |
| | SM + EQL | 0.57 | 0.99 | 0.5829 | 3 | 0.57 | 1.00 | 0.73 | $1.03x_1^4 - 6.01x_1^2 - 1.64x_1 + 0.49x_2^2 - 0.22x_1^5 + 0.64x_1^3 + 0.23x_1x_2$ |

Table 3: Results for XY Model. The true function is $h(x_{p(l)}) = -0.71\cos(x_0 - x_1) - 0.71\cos(x_0 - x_2)$. Estimated terms that do not match the true function are shown in gray. NF = Flow model, DSM = Denoising score model.

| Model | MSE($\downarrow$) | $R^2$($\uparrow$) | RCE($\downarrow$) | ST($\downarrow$) | P($\uparrow$) | R($\uparrow$) | F1($\uparrow$) | Estimated $\hat{h}(x_{p(l)})$ |
|---|---|---|---|---|---|---|---|---|
| MeSSY | - | - | - | - | - | - | - | - |
| NF + PySR | **4.83** | **0.92** | 0.1127 | 0 | 1.00 | 1.00 | 1.00 | $-0.63\cos(0.9x_0 - 0.9x_1) - 0.63\cos(x_0 - x_2)$ |
| DSM + PySR | 42.22 | 0.30 | 0.6200 | 2 | 0.50 | 0.50 | 0.50 | $-0.27\cos(x_0 - x_1) - 0.04x_0x_3 + 0.04x_3^2$ |
| NF + EQL | 6.74 | 0.87 | 0.0141 | 4 | 0.67 | 1.00 | 0.80 | $-0.72\cos(x_0 - x_1) - 0.72\cos(x_0 - x_2) + 0.11\cos(x_0 - x_3) + 0.19\cos^2(x_3) - 0.28\cos(x_0) + 0.32\cos(x_3)$ |
| DSM + EQL | 47.72 | 0.21 | N/A | 2 | 0.33 | 0.50 | 0.40 | $-0.46\cos(0.34x_0 - 0.59x_2 + 0.30x_3 + 0.64\cos(0.48x_0 - 0.61x_2 - 0.33x_3))$ |

performs better than EQL. The quantitative results are summarized in Table 3. In contrast, MeSSY fails to converge and is unable to recover the target expression.

**Scalar $\phi^4$ theory:** We use $10^5$ samples for training flow and score-based models. Here, we train the score-based model with denoising score matching. For SR, the distribution is factorised over patches: a $3 \times 3$ patch is used for estimating $\nabla_{\mathbf{x}}\hat{H}(\mathbf{x})$ via the score model, and a $2 \times 2$ patch for estimating $\hat{H}(\mathbf{x})$ via the flow model. Results are summarised in Table 4.

Both flow and score-based models accurately recover the target energy function, while PySR consistently outperforms EQL. For SR, a $2 \times 2$ patch provides the most effective inductive bias for the $\phi^4$ theory. However, we also assess larger patch sizes, namely $3 \times 3$ and $4 \times 4$, to evaluate the robustness of the method when the interaction range is not known a priori. The corresponding results for $\lambda_1 = 4$ and $\lambda_2 = 1$ using the

flow model are reported in Table 5. Although larger patches substantially increase the dimensionality of the symbolic search space, both PySR and EQL continue to recover the dominant interaction structure.

Notably, PySR consistently recovers the exact local Hamiltonian across all patch sizes without introducing any spurious interaction terms. Even when provided with larger-than-necessary neighborhoods, the recovered expression retains only the physically relevant nearest-neighbor interactions, indicating that the method is able to suppress unnecessary long-range dependencies rather than merely fitting expressions within a prescribed patch. This behavior provides evidence that the framework can identify the underlying interaction structure from an enlarged hypothesis space.

EQL also recovers the dominant Hamiltonian terms for all patch sizes, but introduces an increasing number of spurious interactions as the patch size grows. For the $2 \times 2$, $3 \times 3$, and $4 \times 4$ patches, EQL produces 4, 4, and 7 additional terms, respectively. Many of these involve variables located farther from the central site (e.g., $x_5$, $x_8$, and $x_9$ for the $4 \times 4$ patch), corresponding to interactions that are absent in the true nearest-neighbor Hamiltonian. This trend suggests that the larger symbolic search space makes EQL more susceptible to fitting irrelevant long-range dependencies, although the coefficients of the physically meaningful terms remain close to their ground-truth values.

Overall, these experiments demonstrate that the proposed framework can distinguish relevant from irrelevant interactions even when provided with a larger-than-necessary local neighborhood. In practical applications where the true interaction range may be unknown, larger patches can therefore be employed without substantially affecting recovery accuracy, particularly when using PySR. In contrast, MeSSY fails to converge and is unable to recover the target expression.

Table 4: Results for $\phi^4$ theory with various $\lambda_1, \lambda_2$ settings. The true expressions are: (A) for $\lambda_1 = 4, \lambda_2 = -4$, $h(x_{p(l)}) = 4x_0^4 - 2x_0x_1 - 2x_0x_2$, and (B) for $\lambda_1 = 4, \lambda_2 = 1$, $h(x_{p(l)}) = 4x_0^4 + 5x_0^2 - 2x_0x_1 - 2x_0x_2$. Incorrect terms are highlighted in gray. NF = Flow model, DSM = Denoising score model.

| | Model | MSE($\downarrow$) | $R^2$($\uparrow$) | RCE($\downarrow$) | ST($\downarrow$) | P($\uparrow$) | R($\uparrow$) | F1($\uparrow$) | Estimated $\hat{h}(x_{p(l)})$ |
|---|---|---|---|---|---|---|---|---|---|
| (A) | MeSSY | - | - | - | - | - | - | - | - |
| | NF + PySR | 0.02 | 0.99 | 0.0175 | 0 | 1.00 | 1.00 | 1.00 | $3.91x_0^4 - 1.97x_0x_1 - 1.97x_0x_2$ |
| | NF + EQL | 0.29 | 0.99 | 0.1583 | 4 | 0.43 | 1.00 | 0.60 | $3.42x_0^4 - 2.33x_0x_1 - 2.33x_0x_2 + 0.29x_0x_1^3 + 0.40x_1^3x_3 - 0.13x_0x_3 + 0.49x_1x_2$ |
| | DSM + PySR | 0.01 | 0.99 | 0.0108 | 0 | 1.00 | 1.00 | 1.00 | $3.95x_0^4 - 1.98x_0x_1 - 1.98x_0x_2$ |
| | DSM + EQL | 0.06 | 0.99 | 0.0358 | 1 | 0.75 | 1.00 | 0.86 | $3.65x_0^4 - 2.01x_0x_1 - 2.03x_0x_2 + 0.12x_0^2$ |
| (B) | MeSSY | - | - | - | - | - | - | - | - |
| | NF + PySR | 0.00 | 0.99 | 0.0241 | 0 | 1.00 | 1.00 | 1.00 | $4.01x_0^4 + 5.17x_0^2 - 2.06x_0x_1 - 2.06x_0x_2$ |
| | NF + EQL | 0.07 | 0.99 | 0.0576 | 4 | 0.50 | 1.00 | 0.67 | $3.69x_0^4 + 5.27x_0^2 - 2.01x_0x_2 - 2.14x_0x_1 - 0.68x_1x_2^3 - 0.27x_2^3x_3 + 0.14x_0x_3 + 0.45x_1x_2$ |
| | DSM + PySR | 0.03 | 0.99 | 0.0925 | 0 | 1.00 | 1.00 | 1.00 | $3.12x_0^4 + 5.35x_0^2 - 1.92x_0x_1 - 1.92x_0x_2$ |
| | DSM + EQL | 0.05 | 0.99 | 0.1346 | 5 | 0.44 | 1.00 | 0.62 | $3.07x_0^4 + 5.48x_0^2 - 1.80x_0x_1 - 1.78x_0x_2 - 0.59x_0^3x_1 - 0.54x_0^3x_2 - 0.65x_0^2x_1x_2 - 0.16x_0^2x_1^2 - 0.16x_0^2x_2^2$ |

**Renormalization:** Given the comparable performance of flow-based and score-based models for the $\phi^4$ theory, we employ only the flow model for the renormalisation study. For experiments, we use $10^5$ samples of $64 \times 64$ lattice size. These lattices are progressively coarsened to multiple scales, namely, $d = 32 \times 32$, $16 \times 16$ and $8 \times 8$ by applying $2 \times 2$ average pooling. Separate flow models are trained at each lattice scale. Subsequently, symbolic regression is applied using PySR and EQL, though we report only PySR results due to better convergence and lower mean squared error between $H_\theta(\mathbf{x})$ (from the flow model) and $\hat{H}(\mathbf{x})$ (from SR). A $2 \times 2$ patch is used for lattice factorization.

As the lattice size decreases, perturbative theory predicts a reduction in the coefficient of $x_l^4$ and an increase in the coefficient of $x_l^2$. Our results exhibit the similar trend even in the non-perturbative regime, as shown in Fig. 3. At smaller lattice sizes, the coefficient of $x_l^4$ becomes negligible and is dropped by the method, validating the consistency of the estimated $\hat{H}(\mathbf{x})$.

Table 5: Results for estimating $\hat{h}(x_{p(l)})$ for $\phi^4$ theory with different patch sizes $|p|$ for flow model. True $h(x_{p(l)}) = 4x_0^4 + 5x_0^2 - 2x_0x_1 - 2x_0x_2$. Incorrect terms are shown in gray.

| | $|p|$ | MSE($\downarrow$) | $R^2$($\uparrow$) | RCE($\downarrow$) | ST($\downarrow$) | P($\uparrow$) | R($\uparrow$) | F1($\uparrow$) | Estimated $\hat{h}(x_{p(l)})$ |
|---|---|---|---|---|---|---|---|---|---|
| PySR | $2 \times 2$ | 0.003 | 0.99 | 0.0241 | 0 | 1.00 | 1.00 | 1.00 | $4.01x_0^4 + 5.17x_0^2 - 2.06x_0x_1 - 2.06x_0x_2$ |
| | $3 \times 3$ | 0.006 | 0.99 | 0.0199 | 0 | 1.00 | 1.00 | 1.00 | $4.11x_0^4 + 5.06x_0^2 - 2.04x_0x_1 - 2.04x_0x_2$ |
| | $4 \times 4$ | 0.008 | 0.99 | 0.0158 | 0 | 1.00 | 1.00 | 1.00 | $3.90x_0^4 + 5.14x_0^2 - 1.99x_0x_1 - 1.99x_0x_2$ |
| EQL | $2 \times 2$ | 0.07 | 0.99 | 0.0516 | 4 | 0.50 | 1.00 | 0.67 | $3.69x_0^4 + 5.27x_0^2 - 2.01x_0x_2 - 2.14x_0x_1 - 0.68x_1x_2^3 - 0.27x_2^3x_3 + 0.14x_0x_3 + 0.45x_1x_2$ |
| | $3 \times 3$ | 0.98 | 0.98 | 0.0330 | 4 | 0.50 | 1.00 | 0.67 | $3.86x_0^4 + 5.21x_0^2 - 2.1x_0x_1 - 1.99x_0x_2 + 0.42x_1x_2 - 1.01x_0^3x_2 - 0.14x_0^3x_1 - 0.13x_0^3x_5$ |
| | $4 \times 4$ | 0.11 | 0.99 | 0.0409 | 7 | 0.36 | 1.00 | 0.53 | $3.61x_0^4 + 4.97x_0^2 - 2.09x_0x_1 - 2.03x_0x_2 + 0.11x_0^3x_3 - 0.12x_0^3x_8 - 0.1x_0x_9 + 0.28x_1x_4 + 0.27x_0x_5 + 0.14x_0x_2 + 0.13x_0x_8$ |

To probe potential higher-order interactions, we repeat the analysis using a $3 \times 3$ patch for SR factorization. The resulting action retains the same functional form as that obtained with a $2 \times 2$ patch, indicating the absence of additional long-range and higher-order interaction terms.

To further validate the recovered Hamiltonians beyond coefficient trends, we also perform a quantitative observable-based evaluation. For each lattice scale ($32 \times 32$, $16 \times 16$, and $8 \times 8$), we generate samples from the recovered symbolic Hamiltonians using Hamiltonian Monte Carlo (HMC) and compare them with the corresponding coarse-grained data. We evaluate three physical observables: magnetization ($M$), absolute magnetization ($|M|$), and magnetic susceptibility ($\chi$). The agreement between the coarse-grained and generated distributions is quantified using the percentage overlap between the corresponding histograms and the Earth Mover Distance (EMD) between the observable distributions (Kanaujia et al., 2024). As shown in Table 6, the histogram overlap exceeds 98% for all observables and lattice scales, while the EMD remains very small, ranging from approximately $10^{-4}$ to $10^{-3}$. These results indicate that the recovered Hamiltonians accurately reproduce the observable distributions of the coarse-grained systems.

In addition, we compare the first and second moments of the observables. Table 7 shows that the means and standard deviations computed from samples generated using the recovered Hamiltonians closely match those obtained from the coarse-grained data across all lattice sizes. For example, at lattice size $32 \times 32$, the mean absolute magnetization is 0.0053 for the coarse-grained data and 0.0055 for the generated data, with corresponding standard deviations of 0.0039 and 0.0041, respectively. Similar agreement is observed for magnetization and magnetic susceptibility at all renormalization scales. Overall, these results provide complementary evidence that the recovered symbolic Hamiltonians not only reproduce the qualitative renormalization trends shown in Fig. 3, but also preserve key statistical and physical properties of the coarse-grained system.

Table 6: Quantitative comparison of physical observables computed from coarse-grained data and samples generated from the recovered symbolic Hamiltonians at different renormalization scales. We use 100,000 samples to compute the observables.

| | 32x32 | | 16x16 | | 8x8 | |
|---|---|---|---|---|---|---|
| Observable | % Overlap($\uparrow$) | EMD ($\downarrow$) | % Overlap($\uparrow$) | EMD ($\downarrow$) | % Overlap($\uparrow$) | EMD ($\downarrow$) |
| Magnetization, M | 98.116 | $2.28 \times 10^{-4}$ | 98.845 | $9.09 \times 10^{-5}$ | 98.759 | $8.79 \times 10^{-5}$ |
| Absolute Magnetization, |M| | 98.197 | $2.06 \times 10^{-4}$ | 98.794 | $8.69 \times 10^{-5}$ | 98.749 | $8.30 \times 10^{-5}$ |
| Magnetic Susceptibility | 98.244 | $3.81 \times 10^{-3}$ | 99.187 | $4.47 \times 10^{-4}$ | 99.187 | $1.13 \times 10^{-4}$ |

## 5 Discussion

**Error propagation in two-stage learning:** The proposed framework includes learning at two stages, first the deep generative model, and then, the SR model. Previous works as in Cranmer et al. (2020) report

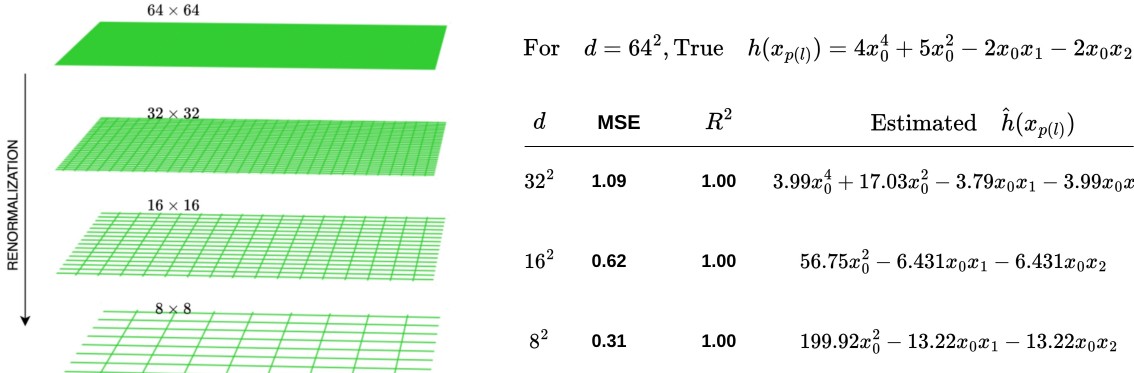

For $d = 64^2$, True $\quad h(x_{p(l)}) = 4x_0^4 + 5x_0^2 - 2x_0x_1 - 2x_0x_2$

| $d$ | MSE | $R^2$ | Estimated $\quad \hat{h}(x_{p(l)})$ |
|---|---|---|---|
| $32^2$ | 1.09 | 1.00 | $3.99x_0^4 + 17.03x_0^2 - 3.79x_0x_1 - 3.99x_0x_2$ |
| $16^2$ | 0.62 | 1.00 | $56.75x_0^2 - 6.431x_0x_1 - 6.431x_0x_2$ |
| $8^2$ | 0.31 | 1.00 | $199.92x_0^2 - 13.22x_0x_1 - 13.22x_0x_2$ |

Figure 3: Renormalization results for non-perturbative region, $\lambda_1 = 4$ and $\lambda_2 = 1$.

Table 7: Observable statistics of the coarse-grained data and generated samples. Mean and standard deviation are reported for magnetization, absolute magnetization, and magnetic susceptibility across different lattice sizes.

| | | Magnetization, M | | Absolute Magnetization, |M| | | Magnetic Susceptibility | |
|---|---|---|---|---|---|---|---|
| | | Mean | Std Dev ($\sigma$) | Mean | Std Dev ($\sigma$) | Mean | Std Dev ($\sigma$) |
| $32 \times 32$ | Coarsened Data | $-1.82 \times 10^{-6}$ | 0.0066 | 0.0053 | 0.0039 | 0.0441 | 0.0609 |
| | Generated Data | $9.09 \times 10^{-5}$ | 0.0068 | 0.0055 | 0.0041 | 0.0481 | 0.0681 |
| $16 \times 16$ | Coarsened Data | $-1.82 \times 10^{-6}$ | 0.0066 | 0.0053 | 0.0039 | 0.0110 | 0.0152 |
| | Generated Data | $-3.36 \times 10^{-5}$ | 0.0067 | 0.0053 | 0.0040 | 0.0115 | 0.0161 |
| $8 \times 8$ | Coarsened Data | $-1.82 \times 10^{-6}$ | 0.0066 | 0.0053 | 0.0039 | 0.0028 | 0.0038 |
| | Generated Data | $-2.31 \times 10^{-5}$ | 0.0067 | 0.0053 | 0.0040 | 0.0029 | 0.0041 |

that using SR alone performs much worse than using it in conjunction with deep models for large scale learning. We find the same with our attempts to apply SR directly to maximise the likelihood. Here, we investigate how errors propagate through these two stages in our framework. Table 8 compares the MSE after flow model and that after the SR stage for various datasets. We use mean subtraction to eliminate the constant term because of normalizing constant $Z$. We see that after SR, with either PySR or EQL, the MSE is less as compared to the MSE just after the flow model. A plausible explanation could be that SR, by its low complexity estimates, reduces the noise in the likelihood information obtained from the flow model. These results further support the use of SR methods instead of only black box deep learning based models for generating samples.

The performance of the proposed framework is primarily governed by the accuracy of the underlying generative model, the extent to which the available data covers the relevant state space, and the scalability of the symbolic regression (SR) algorithm. Since the SR stage operates on densities or score fields estimated by a generative model, inaccuracies in the learned density are directly propagated to the recovered symbolic expression. This effect becomes particularly pronounced for highly multimodal distributions, where normalizing flows exhibit mode-covering behavior (Kanaujia et al., 2024; Kanaujia & Arora, 2025), leading to biased density estimates and, consequently, larger coefficient errors and spurious symbolic terms in the recovered Hamiltonian.

Adequate sampling coverage is equally important for accurate symbolic recovery. The framework assumes that the available samples sufficiently represent all regions that contribute significantly to the target distribution. In systems containing many metastable states separated by energy barriers, obtaining representative samples from all relevant regions can be challenging. As a result, both density estimation and symbolic recovery may deteriorate when important regions of the state space are under-sampled. This phenomenon is evident in the MW-64 benchmark, which combines high dimensionality with strong multimodality and

Table 8: Comparison of $\text{MSE}(H, H_\theta)$, where $H_\theta$ is computed by flow model, and $\text{MSE}(H, \hat{H})$, where $\hat{H}$ is estimated by SR methods (PySR and EQL) using flow model outputs.

| Dataset | $\text{MSE}(H, H_\theta)$ | $\text{MSE}(H, \hat{H})$ | |
|---|---|---|---|
| | | PySR | EQL |
| Many Well, $N = 2$ | 0.01 | 0.00 | 0.00 |
| Many Well, $N = 4$ | 0.05 | 0.00 | 0.00 |
| Many Well, $N = 8$ | 1.60 | 0.00 | 0.23 |
| XY; $\lambda = 1.4^{-1}$ | 9.79 | 4.83 | 6.74 |
| $\phi^4; \lambda_1 = 4, \lambda_2 = -4$ | 1.43 | 0.02 | 0.29 |
| $\phi^4; \lambda_1 = 4, \lambda_2 = 1$ | 0.22 | 0.00 | 0.07 |

contains approximately $2^{32}$ metastable modes. The resulting difficulty in accurately learning the underlying distribution leads to reduced symbolic recovery accuracy.

A second limitation arises from the scalability of the SR stage. While the generative models can often accommodate high-dimensional inputs, the search space explored by SR grows rapidly with the number of variables. This challenge becomes particularly important in systems with long-range interactions, where larger patches are required to capture all relevant interaction terms. Increasing the patch size substantially enlarges the symbolic search space, making exact recovery more difficult and increasing the likelihood of coefficient inaccuracies and spurious terms. Consequently, the overall scalability of the framework is often constrained by the SR component rather than the generative modeling stage.

An additional limitation of the framework is its reliance on domain knowledge in specifying the symbolic library and patch structure. The proposed approach is not intended as a fully automated scientific discovery system; rather, it employs physically motivated inductive biases to make symbolic recovery tractable in high-dimensional settings. Consequently, the quality of the recovered expressions depends on the expressiveness of the chosen symbolic library and the suitability of the patch decomposition for the underlying system. If important functional forms or interaction patterns are excluded from the search space, the recovered symbolic expressions may only provide approximate descriptions of the target Hamiltonian. Accordingly, the recovered expressions should be interpreted as compact and interpretable symbolic approximations of effective actions or Hamiltonians, rather than exact representations of the underlying physical system.

The score-based XY experiments illustrate an additional challenge associated with non-Euclidean state spaces. The XY model evolves on the periodic manifold $(S^1)^N$, whereas standard denoising score matching employs Gaussian perturbations and Euclidean score fields. This mismatch does not fully respect the circular topology of the underlying state space and can lead to larger score estimation errors near periodic boundaries and in regions containing highly oscillatory trigonometric interactions. Since the Hamiltonian is recovered by integrating the estimated score field, these errors may accumulate and reduce symbolic recovery accuracy. Overall, the framework performs best when the generative model accurately represents the target distribution, the available samples provide sufficient coverage of the relevant state space, and the local Hamiltonian can be represented using a moderate number of variables. In more challenging settings, the recovered expressions should be viewed as compact and interpretable symbolic approximations of effective actions or Hamiltonians rather than exact symbolic recoveries of the true underlying dynamics.

We also conduct an extensive analysis of the robustness, repeatability, and overall reproducibility of the proposed framework. These aspects are examined thoroughly through a series of experiments presented in Appendix F. In addition, we also investigate the impact of various hyperparameters on the performance of the method, with a detailed discussion provided in Appendix D.

# 6    Conclusion

We present a general framework for using deep generative models and symbolic regression to estimate equations for energy or Hamiltonian of the system from samples. We demonstrate the use of both likelihood-based and score-based generative models in conjunction with symbolic regression while utilizing density factorization and sample perturbation. Experiments support the effectiveness of the proposed approach in reliably estimating the expressions for the Hamiltonian or action of the XY model and the $\phi^4$ theory, as well as density functions for several popular toy distributions. As an important contribution towards the challenges theorists face in discretizing continuous theories, we demonstrate the ability of the proposed framework to recover compact symbolic approximations of effective actions from discrete samples of $\phi^4$ theory. Furthermore, the renormalization experiments yield symbolic representations in non-perturbative regimes that remain consistent with the coarse-grained dynamics across different scales. An important limitation of the current work stems from the inability of existing SR methods to deal with very large numbers of variables. This hinders the approach in discovering long-range or non-local interaction terms in lattice models, as they may require very large patches for factorization. More generally, the framework relies on physically motivated inductive biases through the choice of symbolic libraries and patch structures, and the quality of recovery depends on the expressiveness of these choices. Nevertheless, our experiments yield good results with patches of size up to $4 \times 4$. Future work includes studying how the mode-covering behavior of deep generative models affects symbolic recovery, and extending the proposed framework to non-equilibrium systems, which evolve dynamically.

# 7    Impact Statement

The potential of this work extends to many areas in computational sciences, including Physics, chemistry and climate science. A common concern for traditional scientists is the inability of machine learning based approaches to give insights into the phenomenon under study. The proposed method gives them both the ability to analyze a large amount of data with machine learning and that to estimate the symbolic equations. The paper includes examples of systems where analytic derivation of equations is not feasible traditionally, but is made feasible by the presented approach. We believe this work can be taken up for follow up studies in the following areas:

- Condensed matter Physics to study phase transitions and renormalization theories
- Nuclear Physics to derive interaction potentials from scattering data
- Climate sciences to model extreme climatic conditions from weather data

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

## A    Data Generation

In this section, we briefly describe the methods used to generate samples from the distributions considered in this study.

### A.1    Multivariate Gaussian Distribution

We use the 'torch.distributions' library to generate samples for training and testing the various models. With a Gaussian distribution having a mean of $[-1.5, 1.5]$ and unit covariance, we generate 25,000 samples for training, 5,000 for validation, and 5,000 for testing the NF, Score, and SR models. Additionally, another set of 20,000 samples is used for inference with the SR model to compute metrics. Similarly, for a Gaussian distribution with mean $[-1.5, 1.5]$ and full covariance matrix $\begin{bmatrix} 1.0 & 0.5 \\ 0.5 & 1.0 \end{bmatrix}$, the same number of samples are generated as detailed above for the earlier setting.

### A.2    Many-Well Distribution

We generate samples from the Many-Well (MW) distribution using rejection sampling as detailed in Midgley et al. (2023). We explore six variants of the Many-Well distribution with varying dimensions (2, 4, 8, 16, 32, and 64). For each variant, we generate 100,000 samples for training, 10,000 for validation, and 10,000 for testing the models. Furthermore, an independent set of 20,000 samples is used for inference with the SR model to compute metrics.

### A.3    XY Model distribution

We use the Metropolis-Hastings (MH) algorithm (Chib & Greenberg, 1995) to generate square lattices of size $8 \times 8$ (64 dim) for the XY model. We set $\lambda = 1.4^{-1}$ and the coupling constant($J$) is set to unity. For proposal distribution, we use a uniform distribution $q(z)$ in the MH algorithm. On account of burn-in, around 30000 initial samples are discarded from the MCMC chain. After every 320 MCMC steps, a configuration is added to the set to have a minimum correlation across samples in the training data. We generate 100,000 samples for training, 10,000 for validation, and 10,000 for testing various models.

### A.4    Scalar $\phi^4$ theory distribution

We use the HMC algorithm (Neal, 2011) to generate samples from the scalar $\phi^4$ theory distribution. Lattices of size $8 \times 8$ (64 dimensions) are generated. The step size is set to 0.05, with 20 steps in the leapfrog integrator of the HMC algorithm. The first 1,000 samples are discarded on account of thermalization. We generate $\phi^4$ data for two different mass settings $\lambda_2 = \{-4, 1\}$, while keeping the coupling constant $\lambda_1$ fixed at 4. For each setting, we use 100,000 samples for training, 10,000 for validation, and 10,000 samples for testing the various models.

For renormalization experiments in scalar $\phi^4$ field theory, we generate $64 \times 64$ (4096-dim) lattices using the HMC algorithm. The simulations are performed in the non-perturbative regime with a coupling constant $\lambda_1 = 4.0$ and mass parameter $\lambda_2 = 1$. The HMC integrator uses a step size of 0.05 and 25 leapfrog steps per trajectory. To ensure thermalization, the first 5000 samples are discarded. From the remaining data, we use 100,000 samples for training and 25,000 samples each for testing and validation.

## B    Model and Training details

In this section, we provide an overview of the model architecture and hyperparameters used in our experiments. We primarily focus on three models. First, we describe the architecture employed in the flow modelling, which is used to compute the negative log-likelihood of the samples. Next, we briefly outline the score estimation models used in the score-based approach. Finally, we discuss the two SR modelling approaches—EQL (Kamienny et al., 2022) and PySR (Cranmer, 2023)—along with their respective training hyperparameters. We run all experiments on four NVIDIA GeForce RTX 3080 GPUs, each with 10 GB of

VRAM. Training time for the PySR model ranged from 10 minutes to 90 minutes, whereas the EQL model required significantly more time, varying between 2 to 10 hours depending on the dataset. While the training time for the flow-based and score-based models ranged from 0.5 to 8 hours, depending on the dataset.

## B.1    Normalizing Flows

We implement the Flow model using the RNVP architecture as detailed in Dinh et al. (2017). For the multivariate Gaussian distribution, we utilize 2 Affine coupling layers for the first variant with unit covariance matrix and 4 coupling layers for the second variant with full covariance matrix, as described in Appendix A.1. Each coupling layer is composed of two dense layers, each containing 256 neurons with ReLU activation. The output consists of two layers: one for scaling and one for translation. The scaling layer outputs two neurons with Tanh activation, while the translation layer outputs two neurons with linear activation. The model is trained with a batch size of 512, using the Adam optimizer with an initial learning rate of $1 \times 10^{-5}$.

For the many-well distribution, we use 10 affine coupling layers to model the distribution across all variants ($n = \{2, 4, 8, 16, 32, 64\}$). The coupling architecture and training details remain the same as described above.

For the XY model dataset, we use 10 affine coupling layers to model the distribution. Each coupling layer consists of 2 convolutional layers with 256 filters of kernel size $3 \times 3$ and ReLU activation, using circular padding. The output layer is made up of 2 components: a scaling layer with Tanh activation and a translation layer with linear activation. The model is trained with a batch size of 128 using the Adam optimizer with an initial learning rate of $1 \times 10^{-5}$.

Since we use a Gaussian distribution as the base distribution for the NF model, which has support over $\mathbf{R}$, the model learns better when the dataset also lies in Euclidean space. However, the XY dataset exists on a circular manifold $\Theta \in [0, 2\pi)^{N \times N}$. To address this, we transform the dataset into Euclidean space using a sigmoid transformation, as detailed in Kanaujia et al. (2024).

For the Scalar $\phi^4$ theory distribution, we use 12 affine coupling layers to model the distribution for both variants. The coupling architecture and training details are the same as those used for the XY dataset, as discussed above.

For the renormalization experiments in scalar $\phi^4$ field theory, we generate lattices of size $64 \times 64$. These are progressively coarsened via average pooling with a $2 \times 2$ filter, reducing the spatial resolution at each step. After the first pooling operation, the lattices are reduced to $32 \times 32$ in size. At this resolution, we model the field distribution using RNVP flow architecture comprising 16 affine coupling layers, each employing 512 convolutional filters. Upon further coarsening, the lattices are downsampled to $16 \times 16$ and subsequently to $8 \times 8$. At the $16 \times 16$ level, we use 12 affine coupling layers, and at $8 \times 8$, we use 10 affine coupling layers. For both of these resolutions, each coupling layer uses 256 convolutional filters. The coupling layer architecture remains consistent across scales, with changes only in the number of filters to accommodate the reduced spatial complexity.

## B.2    Score Model

We estimate the score from the data using a neural network that models the score function $s_\theta(x)$ as followed in Song & Ermon (2019). The model is trained via score matching by minimizing the loss function as described in Eq. (7).

For the multivariate Gaussian distribution, we use a multilayer perceptron (MLP) with the architecture [2, 256, 256, 2] to model the score function. For the many-well distribution, we model the score via an MLP with the architecture [2, 256, 256, 256, 2] for $n = 2$. For $n = \{4, 8, 16, 32, 64\}$, we use an MLP architecture with 4 hidden layers, each consisting of 512 neurons.

For the XY model and scalar $\phi^4$ distributions, we optimize the modeled score function using denoising score matching loss, as described in Eq. (9). The score is modeled with the architecture provided by Song & Ermon (2019). The model, initially downloaded from their GitHub repository, has been adapted to fit the dataset configuration. Training is carried out using the Adam optimizer, beginning with an initial learning rate of $1 \times 10^{-4}$, which is subsequently reduced to $1 \times 10^{-5}$.

### B.3 EQL

This approach builds on the Equation-based Learning (EQL) architecture (Martius & Lampert, 2017; Sahoo et al., 2018; Kim et al., 2021), integrating it with other neural modules to allow end-to-end training via backpropagation. To promote sparse and interpretable solutions, we employ a smoothed variant of the $L_{0.5}$ regularization, which prevents gradient singularities and improves convergence. The regularization function is defined as:

$$
L_{0.5}^*(w) = \begin{cases} |w|^{1/2}, & \text{if } |w| \geq a \\ \left(-\frac{w^4}{8a^3} + \frac{3w^2}{4a} + \frac{3a}{8}\right)^{1/2}, & \text{if } |w| < a \end{cases} \tag{22}
$$

with $a = 0.01$ in all experiments. This penalty is selectively applied to the EQL-specific weights when the model is embedded within a broader neural architecture.

Training is conducted in two phases. The first phase uses a higher learning rate of $10^{-2}$, combined with a regularization weight $\lambda = 5 \times 10^{-3}$ for the multivariate gaussian and many-well distributions and $3 \times 10^{-2}$ for the $\phi^4$ and XY model distributions. This phase spans 20000 iterations, with small regularization weight promoting sparsity in the EQL layers. At the end of this phase, EQL weights with magnitude less than a threshold ($\alpha = 0.01$) are pruned and frozen. In the second phase, training proceeds for 10000 iterations with a reduced learning rate of $10^{-3}$ and no regularization ($\lambda = 0$), enabling fine-tuning of the pruned network.

To balance model expressivity with interpretability, different sets of activation functions are used in the hidden and final layers. For the many-well distribution, we employ a functionally rich set of activations across all layers, including: Constant (1 neuron), Identity (1 neuron), Square $g^2$ (1 neuron), Power 4 $g^4$ (1 neuron), and Multiplication $g_1 \cdot g_2$ (1 neuron). For the multivariate Gaussian distribution, a similar configuration to the many-well distribution is used, with the exception of the Power 4 function, which is omitted. In the case of the $\phi^4$ distribution, the Power 4 and Product functions are applied only in the hidden (forward) layers, while the final layer is limited to Constant, Identity, and Square functions to constrain the complexity of the output. In the score-based modelling of $\phi^4$, the Power 4 function is completely excluded from the primitive set to further reduce the search space and enhance tractability.

For the XY model distribution, which exhibits a circular topology, we introduce an inductive bias by incorporating trigonometric primitives—sin and cos, each with 1 neuron—into both the hidden and final layers. To prevent the nesting of trigonometric functions and reduce symbolic complexity, we adopt a feature transformation strategy in which the input kernel is explicitly expanded to include trigonometric variants. For instance, for a kernel of size 2 with variables $[x_1, x_2, x_3, x_4]$, the input is transformed to $[\cos(x_1), \ldots, \cos(x_4), \sin(x_1), \ldots, \sin(x_4)]$. This approach is similarly extended to other kernel sizes. This transformation not only helps suppress excessive equation complexity but also proves especially beneficial in score-based modelling, as it simplifies integration tasks that would otherwise be intractable.

### B.4 PySR

For the PySR experiments, we use the PySR (Cranmer, 2023) library, which implements a multi-population, tree-based genetic algorithm where multiple evolutions are performed asynchronously. We use PySR with a modified MSE loss, as described in Section 2, under factorization. The experimental parameters are set as follows:

**Sample Size:** We train the model using varying sample sizes based on the convergence of the training loss. For learning the equations with the flow-based and score-based approaches, either 512 or 1024 samples are used, depending on performance. Experiments are conducted with sample sizes of 128, 256, 512, 1024, and 2048. Beyond this range, we observe no further improvement in performance; hence, the aforementioned sample sizes are selected for the experiments.

**Maximum Complexity:** This parameter controls the permissible size of symbolic equations, which corresponds to the number of nodes in the search tree. It is set to 20 for the multivariate Gaussian and many-well

Table 9: EQL configurations and inductive biases used across benchmarks. Numbers in parentheses denote the number of neurons associated with each symbolic operator. The final column indicates whether the primitive library contains the exact functional terms present in the target Hamiltonian.

| Experiment | Operator Library | Layers | Patch Size | Range of Interaction | Exact Terms Present |
|---|---|---|---|---|---|
| Multivariate Gaussian | Const(2), Id(2), Cos(1), Square(2), Exp(2), Mul(2), Pow4(2) | 2/3 | − | − | Yes |
| Many-Well | Const(2), Id(2), Square(3), Exp(2), Mul(1), Pow4(2) | 2/3 | $2 \times 1$ | $2 \times 1$ | Yes |
| $\phi^4$ | Const(1), Id(2), Square(2), Exp(1), Mul(2), Pow4(2) | 2 | $2 \times 2, 3 \times 3, 4 \times 4$ | $2 \times 2$ | Yes |
| XY | Const(1), Id(1), Cos(1), Sin(1), Square(1), Mul(2) | 2 | $2 \times 2, 3 \times 3$ | $2 \times 2$ | Yes |

Table 10: PySR configurations and inductive biases used across benchmarks.

| Experiment | Operator Library | Population | Complexity | Patch Size | Range of Interaction | Exact Terms Present |
|---|---|---|---|---|---|---|
| Multivariate Gaussian | $\{+, -, *\}$ | 10 | 20 | − | − | No |
| Many-Well | $\{+, -, *, \mathrm{pow2}\}$ | 30 | 30 | $2 \times 1$ | $2 \times 1$ | Yes |
| $\phi^4$ | $\{+, -, *, \mathrm{pow2}, \mathrm{pow3}, \mathrm{pow4}\}$ | 30 | 30 | $2 \times 2, 3 \times 3, 4 \times 4$ | $2 \times 2$ | Yes |
| XY | $\{+, -, *, \sin, \cos\}$ | 30 | 30 | $2 \times 2, 3 \times 3$ | $2 \times 2$ | Yes |

distributions, 30 for the $\phi^4$ distribution, and 40 for the XY model dataset. For renormalisation experiments in $\phi^4$ theory, it is set to 30. These values are chosen based on the convergence of the training loss. Increasing the complexity further results in an exponential increase in search time.

**Population Size:** The population size corresponds to the number of candidate solutions evolved in each generation. It is set to 20 for the multivariate Gaussian and many-well distributions and 30 for the $\phi^4$ and the XY model distributions, guided by training loss convergence.

## C  Inductive Biases and Symbolic Search Space.

The symbolic regression stage incorporates domain-specific inductive biases through the choice of primitive operators, expression complexity, and patch size. These design choices constrain the hypothesis space explored by the symbolic regressor and therefore directly influence the class of Hamiltonians that can be recovered. To make these assumptions explicit, Tables 9 and 10 summarize the symbolic regression configurations used across all benchmarks, including the operator library, model complexity, patch size, true interaction range, and whether the primitive set contains the exact functional terms appearing in the target Hamiltonian.

For EQL, symbolic expressions are represented using neural networks whose standard activation functions are replaced with symbolic operators such as identity, multiplication, polynomial, trigonometric, and exponential functions. Expression complexity is controlled through the network depth and the number of neurons assigned to each operator type. The corresponding configurations are reported in Table 9. For PySR, a genetic-programming-based symbolic regression method, the inductive bias is determined by the primitive operator set, maximum expression complexity, and evolutionary population size. The configurations used in our experiments are summarized in Table 10.

It is important to emphasize that these choices incorporate domain knowledge rather than representing unconstrained symbolic discovery. For example, quartic operators are included for the many-well and $\phi^4$ benchmarks because quartic energy terms are known to be physically relevant, multiplicative operators are required to represent interaction terms, and trigonometric operators are included for the XY model because its Hamiltonian is naturally expressed through cosine interactions. Similarly, the patch sizes were selected to match or exceed the known interaction range, ensuring that all relevant local interactions are accessible to the symbolic regressor. Consequently, the recovered expressions should be interpreted as symbolic discovery within a physically motivated hypothesis space rather than completely assumptions-free equation discovery.

# D  Impact of Hyperparameter Choices on Method Performance

In this section, we discuss the sensitivity of the proposed framework to key hyperparameters governing both the generative models and the symbolic regression components. The hyperparameters of the generative model are selected by monitoring performance on a validation set. Similarly, the hyperparameters of the symbolic-regression (SR) methods are adjusted based on the validation error and the model complexity, both of which should remain small. In practice, PySR exhibits higher sensitivity to the number of training samples and to the complexity parameter, whereas EQL is more sensitive to the learning rate.

**i. Number of samples for SR learning:** EQL requires substantially more training data, typically on the order of 10k–100k samples, than PySR. In PySR, we typically use 500–2000 samples. This is in line with the general observation that genetic algorithms require less training data as compared to neural networks. This number, however, increases with the data dimensionality and with the number of modes in the underlying distribution. For example, achieving low complexity and low MSE for MW-64 requires more samples than for $\Phi^4$ (d = 64), owing to the larger number of modes in MW-64.

**ii. Number of samples for NF or score-model learning:** The performance of both NF and score-based models improves as the training sample size increases. To analyse this effect on the overall framework, we conduct a study on the flow-based pipeline (NF with PySR), trained on the $\phi^4$ distribution $(d = 64)$ using varying numbers of training samples. The corresponding results are summarised in Table 11 below.

Table 11: Effect of sample size on model performance.

| No. of Samples | MSE($\downarrow$) | $R^2(\uparrow)$ | Estimated Equation |
|---|---|---|---|
| 10K | 0.47 | 0.994 | $3.5x_0^4 - 1.85x_0x_1 - 1.85x_0x_2$ |
| 50K | 0.07 | 0.999 | $4.2x_0^4 - 2.04x_0x_1 - 2.04x_0x_2$ |
| 100K | 0.01 | 0.999 | $3.9x_0^4 - 1.95x_0x_1 - 2.00x_0x_2$ |

**iii. Complexity control in PySR and EQL:** For PySR, the complexity is typically set between 10 and 30. In EQL, the model complexity is controlled by the architecture of the network, specifically by adjusting the number of layers and neurons in the primitive set of operators. Deeper networks lead to increased model complexity.

In both PySR and EQL, the method struggles to generate an interpretable equation when model complexity is too high or when the dataset fails to sufficiently capture all modes. In multimodal settings, generative models such as NFs often exhibit mode-covering behaviour. This can lead to inaccuracies in likelihood estimation, which, when propagated to the SR stage, may result in the identification of spurious or extraneous terms.

In PySR, both the dataset size and sample quality play a critical role. By sample quality, we refer to the extent to which the data adequately represent all relevant modes of the distribution. The complexity hyperparameter is equally important: as model complexity increases, the inferred symbolic expressions tend to include additional spurious terms. While increasing the dataset size can initially suppress these extraneous terms, beyond a certain point the complexity constraint becomes the dominant factor determining the number of such terms.

Similarly, in EQL, increasing the dataset size helps suppress additional terms in the inferred equation only up to a certain point. Beyond this threshold, the intrinsic complexity of the model, as well as errors propagated from the generative model (e.g., NF or score-based model), begin to significantly influence the accuracy of the SR. These factors can ultimately lead to the appearance of extra terms in the final symbolic expression.

# E  Metrics

We outline the steps for calculating the overall MSE error and $R^2$ score for the proposed approaches. For each sample $x_i$, the corresponding target value is $H(x_i)$, while the symbolic regressor provides the predicted value $\hat{H}(x_i)$. In the NF-based and score-based formulation, the recovered Hamiltonian is defined only

up to an additive constant. Besides integrating the predicted score field introduces an arbitrary integration constant. Although this constant does not alter the underlying energy landscape or the associated probability distribution, it can artificially affect the MSE and $R^2$ metrics during evaluation. Therefore, before computing the metrics, we remove this offset by subtracting the empirical mean from both the target and predicted Hamiltonians:

- Step1: Compute target mean $\bar{H} = \frac{1}{N} \sum_{i=1}^{N} H(x_i)$

- Step2: Mean normalize target values :

$$\tilde{H}(x_i) = H(x_i) - \bar{H} \tag{23}$$

- Step3: Compute predicted mean $\bar{\hat{H}} = \frac{1}{N} \sum_{i=1}^{N} \hat{H}(x_i)$

- Step4: Mean normalise predicted values:

$$\tilde{\hat{H}}(x_i) = \hat{H}(x_i) - \bar{\hat{H}} \tag{24}$$

- Step5: Compute MSE using mean normalized target values and mean normalized predicted values.

$$MSE = \frac{1}{N} \sum_{i=1}^{N} (\tilde{H}(x_i) - \tilde{\hat{H}}(x_i))^2 \tag{25}$$

Apply the same procedure to calculate the $R^2$ score using the mean normalized target values and the mean normalized predicted values. When the score is provided as input to the model, the symbolic regressor (SR) predicts the expression for $-\nabla_x(H(x))$. Upon integrating this expression, a constant term is introduced. To evaluate the model, remove this constant and use the predicted $\hat{H}(x)$ without the constant. Then, proceed with the same evaluation steps as before.

## F    Robustness Analysis

To ensure that the reported results are not artifacts of individual training runs, we investigate the repeatability of each method by performing multiple trials with distinct random seeds. The corresponding performance metrics are aggregated and presented as seed-wise distributions. This analysis highlights the degree of variability across runs and provides a more comprehensive understanding of the reliability of the reported results.

### F.1    Phi-4 Benchmark

For the scalar $\phi^4$ theory benchmark with $d = 64$ (see Table 4 in the main paper), we report results aggregated over multiple runs with different random seeds. To assess the robustness and stability of symbolic recovery, Tables 12 and 13 provide the symbolic recovery metrics and recovered expressions obtained for individual seed settings using the NF + PySR and NF + EQL approaches, respectively. In addition, we also provide histogram-based results aggregated over multiple runs w.r.t. MSE and $R^2$ metric to illustrate the overall variability and distribution.

**NF + PySR (20 seeds)**

| MSE | $R^2$ | Count |
|---|---|---|
| 0.01–0.10 | $> 0.996$ | 7 |
| 0.10–0.15 | 0.994–0.996 | 8 |
| $> 0.15$ | $< 0.994$ | 5 |

**NF + EQL (10 seeds)**

| MSE | $R^2$ | Count |
|---|---|---|
| 0.001–0.08 | > 0.997 | 5 |
| 0.08–0.15 | 0.994–0.997 | 1 |
| > 0.15 | < 0.994 | 4 |

### F.2 Many-Well Benchmark

For the Many-Well distribution ($n = 64$) (see Table 2 in the main paper), we report score-based results aggregated over multiple runs with different random seeds. Tables 14 and 15 present the symbolic recovery metrics and recovered expressions obtained for individual seed settings using the SM + PySR and SM + EQL approaches, respectively. In addition, we provide histogram-based results aggregated across all runs to illustrate the overall variability and distribution of the recovery performance w.r.t. MSE and $R^2$ metric.

**SM + PySR (18 seeds)**

| MSE | $R^2$ | Count |
|---|---|---|
| 0.2–0.5 | > 0.99 | 5 |
| 0.5–1.0 | 0.98–0.99 | 3 |
| 1.0–4.0 | 0.90–0.98 | 5 |
| > 4.0 | < 0.90 | 5 |

**SM + EQL (8 seeds).**

| MSE | $R^2$ | Count |
|---|---|---|
| 0.35–0.50 | > 0.97 | 3 |
| 0.50–4.0 | 0.90–0.97 | 4 |
| > 4.0 | < 0.90 | 1 |

Overall, these seed-dependent histograms demonstrate that the proposed methods achieve reliable performance across runs, with a consistently strong success rate.

## G Additional Experiments on Long-range Interaction Systems

To assess the applicability of the proposed framework beyond nearest-neighbor systems, we consider a class of physical systems with longer-range interactions. We consider a synthetic distribution inspired by the scalar $\phi^4$ model, where the interaction terms involve next-nearest-neighbor couplings rather than nearest-neighbor interactions. For an $8 \times 8$ lattice, we factorize the Hamiltonian into overlapping $5 \times 5$ patches. The corresponding local Hamiltonian $h(x_{p(l)})$, incorporating long-range interactions, can be expressed in terms of patch elements as

| $x_0$ | $x_1$ | $x_2$ | $x_3$ | $x_4$ |
|---|---|---|---|---|
| $x_5$ | $x_6$ | $x_7$ | $x_8$ | $x_9$ |
| $x_{10}$ | $x_{11}$ | $x_{12}$ | $x_{13}$ | $x_{14}$ |
| $x_{15}$ | $x_{16}$ | $x_{17}$ | $x_{18}$ | $x_{19}$ |
| $x_{20}$ | $x_{21}$ | $x_{22}$ | $x_{23}$ | $x_{24}$ |

The Hamiltonian is factorized as

$$H(x) = \sum_l h(x_{p(l)}), \tag{26}$$

where

$$h(x_{p(l)}) = 4x_{12}^4 + 5x_{12}^2 - x_2 x_{12} - x_{10} x_{12} - x_{12} x_{14} - x_{12} x_{22}. \tag{27}$$

Here, $p(l)$ denotes the set of lattice sites contained in the patch anchored at lattice site $l$.

Table 12: Symbolic recovery results for the $\phi^4$ theory across 20 different random seed settings using NF + PySR. The ground-truth local Hamiltonian for $\lambda_1 = 4$ and $\lambda_2 = 1$ is $h(x_{p(l)}) = 4x_0^4 + 5x_0^2 - 2x_0x_1 - 2x_0x_2$. Incorrect terms are highlighted in gray. NF denotes the flow-based model.

| | MSE($\downarrow$) | $R^2(\uparrow)$ | RCE($\downarrow$) | ST($\downarrow$) | P($\uparrow$) | R($\uparrow$) | F1($\uparrow$) | Estimated Equation $\hat{h}(x_{p(l)})$ |
|---|---|---|---|---|---|---|---|---|
| 1 | 0.00 | 0.999 | 0.024 | 0 | 1.00 | 1.00 | 1.00 | $4.01x_0^4 + 5.17x_0^2 - 2.06x_0x_1 - 2.06x_0x_2$ |
| 2 | 0.02 | 0.999 | 0.063 | 0 | 1.00 | 1.00 | 1.00 | $4.76x_0^4 + 4.76x_0^2 - 1.97x_0x_1 - 2.00x_0x_2$ |
| 3 | 0.01 | 0.999 | 0.136 | 0 | 1.00 | 1.00 | 1.00 | $3.72x_0^4 + 7.27x_0^2 - 2.02x_0x_1 - 2.02x_0x_2$ |
| 4 | 0.04 | 0.998 | 0.075 | 1 | 0.75 | 0.75 | 0.75 | $5.93x_0^2 - 2.04x_0x_1 - 2.04x_0x_2 + 4.21x_0^6$ |
| 5 | 0.11 | 0.996 | 0.042 | 0 | 1.00 | 1.00 | 1.00 | $3.82x_0^4 + 5.01x_0^2 - 2.12x_0x_1 - 2.12x_0x_2$ |
| 6 | 0.05 | 0.998 | 0.016 | 1 | 0.75 | 0.75 | 0.75 | $5.01x_0^2 - 2.01x_0x_1 - 2.08x_0x_2 + 3.65x_0^6$ |
| 7 | 0.11 | 0.996 | 0.186 | 0 | 1.00 | 1.00 | 1.00 | $2.09x_0^4 + 5.88x_0^2 - 2.09x_0x_1 - 2.09x_0x_2$ |
| 8 | 0.12 | 0.995 | 0.178 | 1 | 0.80 | 1.00 | 0.89 | $2.16x_0^4 + 5.87x_0^2 - 2.16x_0x_1 - 2.00x_0x_2 + 0.04x_0$ |
| 9 | 0.19 | 0.992 | 0.150 | 1 | 0.80 | 1.00 | 0.89 | $2.66x_0^4 + 5.84x_0^2 - 2.18x_0x_1 - 2.01x_0x_2 + 0.40x_0x_3$ |
| 10 | 0.43 | 0.982 | 0.129 | 0 | 0.75 | 0.75 | 0.75 | $6.68x_0^2 - 2.05x_0x_1 - 2.05x_0x_2$ |
| 11 | 0.09 | 0.996 | 0.055 | 3 | 0.57 | 1.00 | 0.73 | $4.19x_0^4 + 4.78x_0^2 - 1.87x_0x_1 - 1.87x_0x_2 + x_0^6 - 0.89x_0x_2^3 - 0.89x_0x_1^3$ |
| 12 | 0.12 | 0.995 | 0.137 | 2 | 0.67 | 1.00 | 0.80 | $2.51x_0^4 + 5.76x_0^2 - 1.95x_0x_1 - 2.00x_0x_2 - 0.85x_0^3x_1 - 0.87x_0^3x_2$ |
| 13 | 0.07 | 0.997 | 0.063 | 4 | 0.43 | 0.75 | 0.55 | $5.95x_0^2 - 2.00x_0x_1 - 2.00x_0x_2 + 4.04x_0^6 + 1.71x_0 - 0.83x_1 - 0.83x_2$ |
| 14 | 0.17 | 0.993 | 0.191 | 3 | 0.57 | 1.00 | 0.73 | $2.11x_0^4 + 5.91x_0^2 - 2.11x_0x_1 - 2.11x_0x_2 + 0.20x_0^6 - 0.40x_0x_2^3 - 0.40x_0x_1^3$ |
| 15 | 0.11 | 0.995 | 0.109 | 2 | 0.60 | 0.75 | 0.67 | $5.988x_0^2 - 2.13x_0x_1 - 2.13x_0x_2 + 4.18x_0^6 + 0.41x_1x_2$ |
| 16 | 0.10 | 0.996 | 0.052 | 6 | 0.40 | 1.00 | 0.57 | $4.22x_0^4 + 4.85x_0^2 - 1.92x_0x_1 - 1.83x_0x_2 + x_0^6 - 0.91x_0x_2^3 - 0.87x_0x_1^3 + 0.39x_0x_3 - x_0 + x_1$ |
| 17 | 0.31 | 0.987 | 0.125 | 4 | 0.43 | 0.75 | 0.55 | $6.67x_0^2 - 2.04x_0x_1 - 2.04x_0x_2 - 2.04x_0x_1^{10} + 1.61x_1^{10} - x_1 + x_3$ |
| 18 | 0.16 | 0.993 | 0.162 | 5 | 0.44 | 1.00 | 0.62 | $4.28x_0^4 + 4.81x_0^2 - 1.46x_0x_1 - 1.46x_0x_2 - 2.61x_0^3x_1 - 2.61x_0^2x_1x_3 + 0.39x_0^2x_1^2 + 0.39x_0^2x_2^2 + 0.79x_0x_1^2x_3$ |
| 19 | 0.11 | 0.996 | 0.066 | 6 | 0.40 | 1.00 | 0.57 | $4.22x_0^4 + 5.14x_0^2 - 1.82x_0x_1 - 1.82x_0x_2 + 0.14x_0^2x_1^2 + 0.14x_0^2x_2^2 + 0.27x_1x_2x_0^2 - 1.52x_1x_0^3 - 1.52x_2x_0^3 + 0.33x_1x_2$ |
| 20 | 0.13 | 0.994 | 0.095 | 12 | 0.25 | 1.00 | 0.40 | $4.28x_0^4 + 4.75x_0^2 - 1.74x_0x_1 - 1.74x_0x_2 + x_0^6 - 0.38x_1x_0^5 - 0.38x_2x_0^5 + 0.04x_1^2x_0^4 + 0.04x_2^2x_0^4 + 0.07x_1x_2x_0^4 - 1.63x_1x_0^3 - 1.63x_2x_0^3 + 0.16x_0^2x_1^2 + 0.16x_0^2x_2^2 + 0.31x_1x_2x_0^2 + 0.33x_1x_2$ |

The results, summarized in Table 16 indicate that the proposed framework remains effective when the interaction range extends beyond nearest neighbours, provided that the patch size is chosen to encompass the full interaction neighborhood. In particular, NF + PySR and DSM + PySR recover the correct symbolic structure with perfect precision and recall and no spurious terms, demonstrating that the framework can successfully identify interpretable long-range interaction terms from samples alone. However, when EQL is used as the symbolic regressor in both the NF and DSM pipelines, a significant number of spurious terms are recovered, especially in the NF + EQL pipeline. Although most of the true Hamiltonian terms are identified, as reflected by the relatively high recall values, the presence of numerous incorrect terms leads to poor precision.

In physical systems with long-range interactions, the primary challenge arises not from the generative modeling stage, but from the symbolic regression stage. In the proposed framework, the first step uses either a normalizing flow (NF) or a score-based model to convert the unsupervised density estimation problem into a supervised learning problem by generating suitable input-output pairs. The subsequent symbolic regression (SR) step is then used to recover an interpretable expression for the underlying Hamiltonian.

Table 13: Symbolic recovery results for the $\phi^4$ theory across 5 different random seed settings using NF + EQL. The true expressions is $h(x_{p(l)}) = 4x_0^4 + 5x_0^2 - 2x_0x_1 - 2x_0x_2$. Incorrect terms are highlighted in gray.

| | MSE($\downarrow$) | $R^2(\uparrow)$ | RCE($\downarrow$) | ST($\downarrow$) | P($\uparrow$) | R($\uparrow$) | F1($\uparrow$) | Estimated Equation $\hat{h}(x_{p(l)})$ |
|---|---|---|---|---|---|---|---|---|
| 1 | 0.07 | 0.99 | 0.058 | 4 | 0.50 | 1.00 | 0.67 | $3.69x_0^4 + 5.27x_0^2 - 2.01x_0x_2 - 2.14x_0x_1 - 0.68x_1x_2^3 - 0.27x_2^3x_3 + 0.14x_0x_3 + 0.45x_1x_2$ |
| 2 | 0.054 | 0.998 | 0.067 | 11 | 0.27 | 1.00 | 0.42 | $3.46x_0^4 + 4.78x_0^2 - 2.14x_0x_1 - 2.01x_0x_2 - 0.13x_0x_1^3 - 0.32x_0x_2^3 + \cdots$ Other terms |
| 3 | 0.048 | 0.998 | 0.089 | 17 | 0.19 | 1.00 | 0.32 | $3.32x_0^4 + 5.08x_0^2 - 2.07x_0x_1 - 2.28x_0x_2 + 0.06x_0x_3^3 + 0.23x_0x_1^3 + \cdots$ Other terms |
| 4 | 0.054 | 0.098 | 0.065 | 9 | 0.31 | 1.00 | 0.47 | $3.46x_0^4 + 4.78x_0^2 - 2.14x_0x_1 - 2.04x_0x_2 - 0.32x_0x_2^3 - 0.13x_0x_1^3 + \cdots$ Other terms |
| 5 | 0.041 | 0.998 | 0.066 | 24 | 0.14 | 1.00 | 0.25 | $3.61x_0^4 + 5.06x_0^2 - 2.26x_0x_1 - 2.05x_0x_2 - 0.16x_1^2x_2^2 + 0.28x_0^2x_1^2 + \cdots$ Other terms |

Table 14: Symbolic recovery results for the MW-64 benchmark across 18 random seed settings using SM + PySR. The true local Hamiltonian is $h(x_{p_l}) = x_1^4 - 6x_1^2 - 0.5x_1 + 0.5x_2^2$. Incorrect terms are highlighted in gray.

| | MSE($\downarrow$) | $R^2(\uparrow)$ | RCE($\downarrow$) | ST($\downarrow$) | P($\uparrow$) | R($\uparrow$) | F1($\uparrow$) | Estimated Equation $\hat{h}(x_{p(l)})$ |
|---|---|---|---|---|---|---|---|---|
| 1 | 0.72 | 0.985 | 0.279 | 0 | 1.00 | 1.00 | 1.00 | $1.58x_1^4 - 7.27x_1^2 - 0.65x_1 + 0.51x_2^2$ |
| 2 | 0.46 | 0.990 | 0.126 | 0 | 1.00 | 1.00 | 1.00 | $1.37x_1^4 - 6.64x_1^2 - 0.51x_1 + 0.50x_2^2$ |
| 3 | 0.32 | 0.993 | 0.242 | 1 | 0.80 | 1.00 | 0.89 | $1.68x_1^4 - 7.46x_1^2 - 0.52x_1 + 0.50x_2^2$ |
| 4 | 0.28 | 0.994 | 0.160 | 0 | 1.00 | 1.00 | 1.00 | $1.46x_1^4 - 6.96x_1^2 - 0.50x_1 + 0.51x_2^2$ |
| 5 | 0.40 | 0.991 | 0.179 | 0 | 1.00 | 1.00 | 1.00 | $1.44x_1^2 - 6.92x_1^2 - 0.45x_1 + 0.51x_2^2$ |
| 6 | 0.42 | 0.991 | 0.189 | 0 | 1.00 | 1.00 | 1.00 | $1.49x_1^4 - 7.02x_1^2 - 0.55x_1 + 0.50x_2^2$ |
| 7 | 0.64 | 0.986 | 0.134 | 1 | 0.80 | 1.00 | 0.89 | $1.38x_1^4 - 6.59x_1^2 - 0.47x_1 + 0.50x_2^2 - 0.11x_1^3$ |
| 8 | 0.65 | 0.986 | 0.264 | 1 | 0.75 | 0.75 | 0.75 | $1.58x_1^4 - 7.14x_1^2 + 0.51x_2^2 - 0.28x_1^3$ |
| 9 | 0.70 | 0.985 | 0.242 | 1 | 0.80 | 1.00 | 0.89 | $1.67x_1^4 - 7.50x_1^2 - 0.48x_1 + 0.50x_2^2 - 0.10x_1^3$ |
| 10 | 1.03 | 0.978 | 0.178 | 1 | 0.80 | 1.00 | 0.89 | $1.52x_1^4 - 6.98x_1^2 - 0.50x_1 + 0.51x_2^2 - 0.11x_1^3$ |
| 11 | 1.45 | 0.969 | 0.186 | 0 | 1.00 | 1.00 | 1.00 | $1.31x_1^4 - 6.60x_1^2 - 0.34x_1 + 0.51x_2^2$ |
| 12 | 2.98 | 0.936 | 0.351 | 0 | 1.00 | 1.00 | 1.00 | $1.57x_1^4 - 7.18x_1^2 - 0.81x_1 + 0.51x_2^2$ |
| 13 | 3.48 | 0.925 | 0.214 | 1 | 0.80 | 1.00 | 0.89 | $1.52x_1^4 - 6.98x_1^2 - 0.58x_1 + 0.50x_2^2 - 0.17x_1^3$ |
| 14 | 3.91 | 0.916 | 0.224 | 1 | 0.80 | 1.00 | 0.89 | $1.49x_1^4 - 6.87x_1^2 - 0.63x_1 + 0.50x_2^2 - 0.14x_1^3$ |
| 15 | 10.78 | 0.768 | 0.405 | 1 | 0.80 | 1.00 | 0.89 | $1.64x_1^4 - 7.31x_1^2 - 0.87x_1 + 0.51x_2^2 - 0.14x_1^3$ |
| 16 | 16.66 | 0.641 | 0.645 | 1 | 0.80 | 1.00 | 0.89 | $1.51x_1^4 - 7.01x_1^2 - 1.44x_1 + 0.51x_2^2 - 0.12x_1^3$ |
| 17 | 27.35 | 0.410 | 0.735 | 1 | 0.80 | 1.00 | 0.89 | $1.58x_1^4 - 7.21x_1^2 - 1.58x_1 + 0.50x_2^2 + 0.11x_1^3$ |
| 18 | 42.24 | 0.089 | 0.867 | 1 | 0.80 | 1.00 | 0.89 | $1.43x_1^4 - 6.74x_1^2 - 1.96x_1 + 0.50x_2^2 + 0.20x_1^3$ |

The scalability of the overall pipeline is therefore largely determined by the scalability of the SR method. Most existing symbolic regression algorithms perform well only for a relatively small number of input variables (typically fewer than 10). In systems with long-range interactions, larger patches are required to ensure that all relevant interaction terms are captured within the local Hamiltonian. For example, interactions extending beyond nearest neighbors may require patch sizes of $3 \times 3$, $4 \times 4$, $5 \times 5$, or larger. As the patch size increases, the number of variables presented to the symbolic regressor grows rapidly, significantly enlarging the search space of possible expressions. This generally leads to reduced symbolic recovery accuracy, larger coefficient errors, and an increased number of spurious terms.

The results on synthetic long-range interaction systems show that the proposed framework can still successfully recover the correct interaction structure. However, the difficulty of symbolic recovery increases as the number of variables grows. In particular, we observed that PySR, which is based on genetic programming, remains comparatively robust in these higher-dimensional settings, whereas EQL, a neural-network-based

Table 15: Symbolic recovery results for the MW-64 benchmark across 8 random seed settings using SM + EQL. The true local Hamiltonian is $h(x_{p_l}) = x_1^4 - 6x_1^2 - 0.5x_1 + 0.5x_2^2$. Incorrect terms are highlighted in gray.

| | MSE($\downarrow$) | $R^2(\uparrow)$ | RCE($\downarrow$) | ST($\downarrow$) | P($\uparrow$) | R($\uparrow$) | F1($\uparrow$) | Estimated Equation $\hat{h}(x_{p(l)})$ |
|---|---|---|---|---|---|---|---|---|
| 1 | 0.57 | 0.997 | 0.583 | 3 | 0.57 | 1.00 | 0.73 | $1.03x_1^4 - 6.01x_1^2 - 1.64x_1 + 0.49x_2^2 - 0.22x_1^5 + 0.64x_1^3 + 0.23x_1x_2$ |
| 2 | 0.92 | 0.971 | 1.452 | 2 | 0.67 | 1.00 | 0.80 | $0.89x_1^4 - 5.65x_1^2 + 2.30x_1 + 0.48x_2^2 + 0.14x_1^5 - 1.09x_1^3$ |
| 3 | 1.78 | 0.953 | 0.868 | 4 | 0.50 | 1.00 | 0.67 | $0.51x_1^4 - 5.14x_1^2 + 0.89x_1 + 0.47x_2^2 + 0.16x_1^6 - 0.12x_1^5 - 0.24x_1^3 - 0.14x_1x_2$ |
| 4 | 0.57 | 0.976 | 1.517 | 2 | 0.67 | 1.00 | 0.80 | $0.81x_1^4 - 5.52x_1^2 + 2.38x_1 + 0.48x_2^2 + 0.16x_1^5 - 1.16x_1^3$ |
| 5 | 2.01 | 0.949 | 1.570 | 3 | 0.57 | 1.00 | 0.73 | $0.73x_1^4 - 5.46x_1^2 + 2.44x_1 + 0.48x_2^2 + 0.19x_1^5 - 1.23x_1^3 + 0.13x_1x_2$ |
| 6 | 2.58 | 0.928 | 1.205 | 2 | 0.67 | 1.00 | 0.80 | $1.33x_1^4 - 6.49x_1^2 + 1.69x_1 + 0.48x_2 + 0.13x_1^5 - 0.89x_1^3$ |
| 7 | 0.48 | 0.979 | 0.788 | 3 | 0.57 | 1.00 | 0.73 | $0.18x_1^4 - 4.32x_1^2 + 0.51x_1 + 0.48x_2^2 + 0.17x_1^6 + 0.11x_1^5 - 0.61x_1^3$ |
| 8 | 3.84 | 0.905 | 1.850 | 3 | 0.57 | 1.00 | 0.73 | $1.50x_1^4 - 6.74x_1^2 + 2.87x_1 + 0.48x_2^2 + 0.23x_1^5 - 1.36x_1^3 + 0.18x_1x_2$ |

Table 16: Results for a synthetic long-range interaction system. The true expression is $h(x_{p(l)}) = 4x_{12}^4 + 5x_{12}^2 - x_2x_{12} - x_{10}x_{12} - x_{12}x_{14} - x_{12}x_{22}$ Incorrect terms are highlighted in gray. NF = Flow model, DSM = Denoising score model.

| Method | MSE($\downarrow$) | $R^2(\uparrow)$ | RCE($\downarrow$) | ST($\downarrow$) | P($\uparrow$) | R($\uparrow$) | F1($\uparrow$) | Estimated Equation $\hat{h}(x_{p(l)})$ |
|---|---|---|---|---|---|---|---|---|
| NF + PySR | 0.7711 | 0.9704 | 0.1428 | 0 | 1.00 | 1.00 | 1.00 | $4.73x_{12}^4 + 4.73x_{12}^2 - 0.83x_2x_{12} - 0.84x_{10}x_{12} - 0.85x_{12}x_{14} - 0.86x_{12}x_{22}$ |
| DSM + PySR | 0.8372 | 0.9680 | 0.0570 | 0 | 1.00 | 1.00 | 1.00 | $3.40x_{12}^4 + 5.16x_{12}^2 - 0.96x_2x_{12} - 0.95x_{10}x_{12} - 0.97x_{12}x_{14} - 0.96x_{12}x_{22}$ |
| NF + EQL | 13.8171 | 0.4225 | 1.5385 | 1467 | 0.003 | 1.00 | 0.005 | $0.26x_{12}^4 + 1.62x_{12}^2 + 2.07x_2x_{12} + 1.47x_{10}x_{12} - 2.11x_{12}x_{14} - 1.97x_{12}x_{22} + \cdots$ Higher order terms |
| DSM + EQL | 23.8519 | 0.0031 | 0.9181 | 4 | 0.50 | 0.67 | 0.57 | $0.03x_{12}^4 - 0.12x_2x_{12} - 0.10x_{10}x_{12} - 0.10x_{12}x_{14} - 0.16x_7x_{12} + 0.03x_{11}x_{12} - 0.05x_{12}x_{13} + 0.06x_{12}x_{17}$ |

symbolic regressor, tends to exhibit a more pronounced degradation in performance, often producing additional spurious terms and less accurate coefficient estimates.

Overall, the proposed framework is applicable to systems with long-range interactions, provided that the patch size is large enough to capture the relevant dependencies.

## H   Symbolic Recovery Under Primitive Library Misspecification

To investigate the effect of library misspecification, we first consider the scalar $\phi^4$ benchmark and remove the explicit polynomial basis functions $\{x, x^2, x^3, x^4\}$ from the primitive library, retaining only the basic arithmetic operators $\{+, -, \times\}$ together with the trigonometric operators $\{\sin, \cos\}$.

We note, however, that this setting should not be regarded as a strictly misspecified experiment. Although explicit polynomial basis functions are removed, the target Hamiltonian remains representable because polynomial terms can still be reconstructed through repeated applications of the multiplication operator. We therefore view this experiment as a test of robustness when important basis functions are removed, forcing the symbolic regressor to reconstruct them indirectly, rather than as a true misspecification experiment. More generally, misspecification in symbolic regression is often not absolute, since many functions can be reconstructed or approximated through compositions of other operators. For example, polynomial terms can

Table 17: Symbolic recovery on the scalar $\phi^4$ benchmark under primitive library misspecification. Explicit polynomial primitives are removed from the symbolic library, leaving only arithmetic and trigonometric operators.

| Method | MSE($\downarrow$) | $R^2$($\uparrow$) | Estimated $\hat{h}(x_{p(l)})$ |
|---|---|---|---|
| NF + PySR | 0.0085 | 0.99 | $3.89x_0^4 + 5.15x_0^2 - 2.00x_0x_1 - 2.00x_0x_2$ |
| NF + EQL | 2.2872 | 0.90 | $5.10x_0^2 - 2.16x_0x_1 - 1.75x_0x_2 + 2.62x_0x_3 + 0.25x_1x_2 + \cdots +$ higher-order polynomial terms $+ \cdots +$ nested $\sin, \cos$ terms |

be generated via repeated multiplication, while trigonometric functions admit polynomial approximations through Taylor expansions. Consequently, the distinction between a specified and misspecified library often depends on the complexity of the resulting representation, rather than the strict representability.

Table 17 reports the results obtained using the flow model followed by symbolic regression. Despite the removal of explicit polynomial basis functions from the primitive library, PySR recovers a highly accurate approximation of the target Hamiltonian, achieving an $R^2$ value of 0.99 and recovering the dominant interaction structure. In contrast, EQL exhibits a noticeable degradation in the performance and introduces additional spurious terms involving higher-order polynomial combinations and nested trigonometric functions. These results indicate that symbolic recovery becomes substantially more challenging when the symbolic hypothesis space (primitive set) is not well aligned with the true functional form. Even when the target Hamiltonian remains representable, recovery quality depends not only on the primitive set but also on the ability of the symbolic regressor to efficiently reconstruct the target expression from the available primitives.

### H.1 Stronger Misspecification: XY Model Without Trigonometric Operators

To construct a more stringent misspecification setting, we consider the XY model, whose ground-truth local Hamiltonian is

$$h(x_{p(l)}) = -0.71\cos(x_0 - x_1) - 0.71\cos(x_0 - x_2). \tag{28}$$

Unlike the $\phi^4$ experiment above, we remove all trigonometric operators (sin and cos) from the primitive library and retain only arithmetic operators. In this setting, the target interaction is no longer directly representable within the primitive set, and the symbolic regressor must approximate the cosine interaction indirectly through polynomial compositions.

Using the Taylor expansion

$$\cos(z) = 1 - \frac{z^2}{2!} + \frac{z^4}{4!} - \cdots, \tag{29}$$

the XY Hamiltonian can be approximated by a polynomial containing quadratic, quartic, and mixed interaction terms. Consequently, successful recovery in this setting requires the symbolic regressor to construct an indirect polynomial approximation to the underlying trigonometric interaction.

$$\begin{aligned} h(x_{p(l)}) &\approx -0.71\left(1 - \frac{(x_0 - x_1)^2}{2!} + \frac{(x_0 - x_1)^4}{4!} + \cdots\right) - 0.71\left(1 - \frac{(x_0 - x_2)^2}{2!} + \frac{(x_0 - x_2)^4}{4!} + \cdots\right) \\ &\approx -1.42 + 0.71x_0^2 + 0.36x_1^2 + 0.36x_2^2 - 0.71x_0x_1 - 0.71x_0x_2 \\ &\quad + 0.06x_0^4 + 0.03x_1^4 + 0.03x_2^4 - 0.12x_0^3x_1 - 0.12x_0^3x_2 \\ &\quad + 0.12x_0x_1^3 - 0.12x_0x_2^3 + 0.18x_0^2x_1^2 + 0.18x_0^2x_2^2 + \cdots \end{aligned} \tag{30}$$

Table 18: Symbolic recovery results for the XY model under primitive library misspecification. The symbolic regressor approximates the cosine interaction indirectly through polynomial compositions.

| Method | MSE($\downarrow$) | $R^2(\uparrow)$ | Estimated $\hat{h}(x_{p(l)})$ |
|---|---|---|---|
| NF + PySR | 8.84 | 0.81 | $-0.03x_0^2x_1^2-0.03x_0^2x_2^2+0.03x_0x_1^3+0.02x_0x_2^3+0.02x_0^3x_2+0.15x_0^2+0.56x_1^2-$ $0.43x_0x_1 - 0.30x_0x_2 - 0.07x_1^3 - 0.07x_0x_1^2 + 0.15x_0^2x_1 + 0.15x_1$ |
| NF + EQL | 7.13 | 0.85 | $0.46x_0^2 + 0.14x_1^2 + 0.17x_2^2 - 0.37x_0x_1 - 0.24x_0x_2 + 0.14x_1 + 0.11x_0$ |

The results are reported in Table 18. Despite the absence of trigonometric operators, both PySR and EQL recover meaningful approximations to the underlying interaction, achieving $R^2$ values of 0.81 and 0.85, respectively. The recovered expressions primarily consist of low-order polynomial interaction terms, consistent with the expected Taylor-series approximation of the cosine function.

Compared to the $\phi^4$ experiment, this setting constitutes a substantially stronger form of misspecification because the true functional form is absent from the primitive library. Nevertheless, the quantitative results in Table 18 show that both symbolic regression methods are able to recover meaningful approximations of the underlying cosine interaction. Although the recovered expressions do not match the true Hamiltonian exactly, they capture several dominant low-order terms expected from the Taylor expansion of the cosine function, including quadratic self-interaction terms and pairwise interactions such as $x_0x_1$ and $x_0x_2$. The resulting approximations achieve reasonably high coefficients of determination ($R^2 = 0.81$ for PySR and $R^2 = 0.85$ for EQL), indicating that a substantial fraction of the Hamiltonian variance is explained even under strong library misspecification. Interestingly, EQL attains a slightly higher $R^2$ and lower MSE than PySR in this setting, suggesting that lower-order polynomial approximations may provide a more robust surrogate when the true trigonometric functional form is unavailable. Overall, these results indicate that the proposed framework can still recover useful symbolic approximations when the target Hamiltonian is not directly representable, although with a noticeable degradation in accuracy relative to the well-specified setting. More broadly, they highlight that symbolic recovery quality depends not only on primitive library but also on the ability of the symbolic regressor to construct effective approximations within the restricted hypothesis space.

Taken together, the $\phi^4$ and XY experiments highlight an important aspect of the framework: successful symbolic recovery depends not only on the quality of the learned generative model but also on the expressiveness of the primitive library. When the target functional form is not directly available within the primitive set, the symbolic regressor may still recover useful approximations through compositions of simpler operators, albeit with increased complexity and reduced accuracy. Consequently, the recovered expressions should be interpreted as effective symbolic approximations rather than exact recoveries of the underlying Hamiltonian.

## H.2 Error Propagation Under Primitive Set Misspecification

A central claim of the proposed framework is that symbolic regression can act as a denoising or compression step by reducing the approximation error present in the learned generative model. However, when important basis functions are removed from the primitive library, it is not obvious whether this error-reduction property continues to hold.

To investigate this question, we compare the error of the learned flow model, $\text{MSE}(H, H_\theta)$, against the error of the recovered symbolic Hamiltonian, $\text{MSE}(H, \hat{H})$, for the restricted-library experiments described above. The results are reported in Table 19.

The results demonstrate that error reduction is not automatic and depends on the ability of the symbolic regressor to construct an accurate approximation within the available hypothesis space. In the $\phi^4$ experiment, PySR substantially reduces the error relative to the underlying flow model despite the removal of explicit polynomial basis functions, reducing the MSE from 0.22 to 0.01. This indicates that PySR is able to efficiently reconstruct the missing polynomial structure through compositions of the remaining primitives. In contrast, EQL introduces several additional spurious terms, resulting in an error substantially larger than that of the underlying flow model.

Table 19: Error propagation analysis under primitive library misspecification. $\mathrm{MSE}(H, H_\theta)$ denotes the error of the learned flow model relative to the true Hamiltonian, while $\mathrm{MSE}(H, \hat{H})$ denotes the error of the recovered symbolic approximation.

|  | Method | $\mathrm{MSE}(H, H_\theta)$ | $\mathrm{MSE}(H, \hat{H})$ |
|---|---|---|---|
| $\phi^4; \lambda_1 = 4, \lambda_2 = 1$ | NF + PySR | 0.22 | 0.01 |
|  | NF + EQL |  | 2.29 |
| XY; $\lambda = 1.4^{-1}$ | NF + PySR | 9.79 | 8.84 |
|  | NF + EQL |  | 7.13 |

A different behavior is observed in the XY experiment, where all trigonometric operators are removed from the primitive library. In this setting, the target cosine interaction is no longer directly representable and must be approximated indirectly through polynomial compositions. Both PySR and EQL reduce the error relative to the learned flow model, with EQL achieving a lower symbolic approximation error. Nevertheless, the reduction is considerably smaller compared to the $\phi^4$ experiment, reflecting the increased difficulty in approximating the functional form.

In general, these results indicate that the error-reduction property of symbolic regression is strongly dependent on the degree of misspecification in the primitive set and the expressive power of the underlying SR algorithm. When the target Hamiltonian remains representable, symbolic regression can substantially reduce modeling errors by recovering compact symbolic representations. Under stronger misspecification, where the true functional form is absent from the primitive set, symbolic regression can still yield meaningful approximations, although the achievable error reduction is more limited.

