# OpenReview forum: "Symbolic Density Estimators for Unnormalized Distributions"
_TMLR — Under review for TMLR_

### Review · Reviewer_hHfF · 2026-04-14

**Summary Of Contributions:**

This paper proposed a framework for recovering symbolic expressions of unnormalized densities based on a collection of samples from the target density. The main idea is based on combining existing techniques from deep generative modeling and symbolic regression, which leads to a two-stage framework. Numerical experiments on toy models and distributions from models in computational physics, such as 2D Gaussian densities, many-well distributions, XY model and scalable $\phi^4$ field theory are provided to justify the validity of the proposed methodology.

**Additional Comments:**

NA

**Audience:**

Yes

**Audience Explanation:**

To the best of the reviewer’s knowledge, AI for Science, Scientific Machine Learning, and generative modeling are very active research areas recently. Given that the approach proposed in this paper provides a clear way of obtaining interpretable symbolic expressions for unnormalized densities in physics, the reviewer expects that TMLR readers will be interested in the paper’s results.

**Claims And Evidence:**

Yes

**Claims Explanation:**

My answer to the question above is somewhat a mixture of "Yes" and "No". Specifically, the authors have provided numerical evidence to show that the recovered symbolic formulas are close to the ground truth under metrics like MSE and $R^2$. However, since the paper's main focus/selling point is on the the interpretability of the proposed method and its application in scientific discovery, the reviewer thinks that it might be necessary for the authors to consider adding a few more challenging cases in the numerical experiments. For instance, would it be possible for the authors to comment on how the method might on physical systems with long-range interactions? The reviewer thinks that such extra results will definitely make the paper more convincing.

**Requested Changes:**

In addition to the extra experiments described above, the reviewer thinks that it might also be necessary for the authors to revise on the manuscript's presentation. Specifically, the authors are encouraged to consider adding a subsection to describe the basics of symbolic regression with mathematical formulas. As far as the reviewer knows, this is not obvious to everyone in the general ML community. Therefore, adding such a short description might enhance the exposure of the manuscript.

---

> ### Author Response · Authors · 2026-06-11
>
> >We thank the reviewer for this insightful suggestion. We agree that, beyond predictive accuracy metrics such as MSE and $R^2$, evaluating the symbolic structure of the recovered expressions is essential for assessing interpretability and scientific utility. In response, we have expanded the experimental evaluation to include additional symbolic recovery metrics. Specifically, we report the **relative coefficient error**, which measures the deviation of recovered coefficients from the ground-truth coefficients for matched terms; the **number of spurious terms**, which quantifies the number of recovered terms absent from the true expression; and **precision**, **recall**, and **F1 score**, which evaluate the accuracy of symbolic term recovery. Precision measures the fraction of recovered terms that are physically correct, recall measures the fraction of true terms successfully recovered, and the F1 score provides a balanced summary of both. Together, these metrics provide a more direct assessment of interpretability and symbolic recovery than MSE and $R^2$ alone.
> >
> >In response to the reviewer's suggestion regarding more challenging physical systems, we have additionally included experiments involving longer-range interactions. We consider a synthetic distribution inspired by the scalar $\phi^4$ model, where the interaction terms involve next-nearest-neighbor couplings rather than nearest-neighbor interactions. For an $8 \times 8$ lattice, we factorize the Hamiltonian into overlapping $5 \times 5$ patches as shown below. The corresponding local Hamiltonian $h(x_{p(l)})$  incorporating long range interactions  in terms of patch elements  can be expressed as
> >
> >
> >|          |          |          |          |          |
> >|----------|----------|----------|----------|----------|
> >| $x_0$    | $x_1$    | $x_2$    | $x_3$    | $x_4$    |
> >| $x_5$    | $x_6$    | $x_7$    | $x_8$    | $x_9$    |
> >| $x_{10}$ | $x_{11}$ | $x_{12}$ | $x_{13}$ | $x_{14}$ |
> >| $x_{15}$ | $x_{16}$ | $x_{17}$ | $x_{18}$ | $x_{19}$ |
> >| $x_{20}$ | $x_{21}$ | $x_{22}$ | $x_{23}$ | $x_{24}$ |
> >|          |          |          |          |          |
> >
> > $H(x)= \sum_{l} h(x_{p(l)})$,
> >
> >$h(x_{p(l)})= 4x_{12}^4 + 5x_{12}^2 -x_{2}x_{12} -x_{10}x_{12}-x_{12}x_{14}-x_{12}x_{22}$
> >
> >Where $p(l)$ is the set of lattice sites anchored in the patch anchored at lattice site $l$.
> >
> >
> >|            | MSE($\downarrow$) | $R^2(\uparrow)$ | Relative Coefficient Error | # Spurious Terms | Precision | Recall | F1 Score | Estimated $\hat h(x_{p(l)})$                                                                                                           |   |
> >|------------|-------------------|-----------------|----------------------------|------------------|-----------|--------|----------|----------------------------------------------------------------------------------------------------------------------------------------|---|
> >| NF + PySR  | 0.7711            | 0.9704          | 0.1428                     | 0                | 1.00      | 1.00   | 1.00     | $4.73x_{12}^4 + 4.73x_{12}^2- 0.83x_{2}x_{12}-0.84x_{10}x_{12}-0.85x_{12}x_{14}-0.86x_{12}x_{22}$                                      |   |
> >| DSM + PySR | 0.8372            | 0.9680          | 0.0570                     | 0                | 1.00      | 1.00   | 1.00     | $3.40x_{12}^4 + 5.16x_{12}^2- 0.96x_{2}x_{12}-0.95x_{10}x_{12}-0.97x_{12}x_{14}-0.96x_{12}x_{22}$                                      |   |
> >| NF + EQL   | 13.8171           | 0.4225          | 0.7942                     | 1459             | 0.004     | 1.00   | 0.008    | $0.18x_{12}^4 + 3.05x_{12}^2- 0.17x_{2}x_{12}-1.99x_{10}x_{12}-2.08x_{12}x_{14}-0.48x_{12}x_{22}+....$ Other  higher order terms       |   |
> >| DSM + EQL  | 23.8519           | 0.0031          | 0.9181                     | 4                | 0.50      | 0.67   | 0.57     | $0.03x_{12}^4 - 0.12x_{2}x_{12}-0.16x_{7}x_{12}-0.10x_{10}x_{12} +0.03x_{11}x_{12}-0.05x_{12}x_{13}-0.10x_{12}x_{14}+0.06x_{12}x_{17}$ |   |
> >
> >These results indicate that the proposed framework remains effective when the interaction range extends beyond nearest neighbors, provided that the patch size is chosen to encompass the full interaction neighborhood. In particular, NF + PySR and DSM + PySR recover the correct symbolic structure with perfect precision and recall and no spurious terms, demonstrating that the framework can successfully identify interpretable long-range interaction terms from samples alone. However, when EQL is used as the symbolic regressor in both the NF and DSM pipelines, a significant number of spurious terms are recovered especially in NF + EQL pipeline. Although most of the true Hamiltonian terms are identified, as reflected by the relatively high recall values, the presence of numerous incorrect terms leads to poor precision.

---

> ### Author Response · Authors · 2026-06-11
>
> For instance, would it be possible for the authors to comment on how the method might on physical systems with long-range interactions?
>
> >In physical systems with long-range interactions, the primary challenge arises not from the generative modeling stage, but from the symbolic regression stage. In our framework, the first step uses either a normalizing flow (NF) or a score-based model to convert the unsupervised density estimation problem into a supervised learning problem by generating suitable input-output pairs. The subsequent symbolic regression (SR) step is then used to recover an interpretable expression for the underlying Hamiltonian.
> >
> >------
> >
> >The scalability of the overall pipeline is therefore largely determined by the scalability of the SR method. Most existing symbolic regression algorithms perform well only for a relatively small number of input variables (typically fewer than 10). In systems with long-range interactions, larger patches are required to ensure that all relevant interaction terms are captured within the local Hamiltonian. For example, interactions extending beyond nearest neighbors may require patch sizes of $3\times3$, $4\times4$, $5\times5$, or larger. As the patch size increases, the number of variables presented to the symbolic regressor grows rapidly, significantly enlarging the search space of possible expressions. This generally leads to reduced symbolic recovery accuracy, larger coefficient errors, and an increased number of spurious terms.
> >
> >------------
> >
> >The results on synthetic long-range interaction systems show that the proposed framework can still successfully recover the correct interaction structure. However, the difficulty of symbolic recovery increases as the number of variables grows. In particular, we observed that PySR, which is based on genetic programming, remains comparatively robust in these higher-dimensional settings, whereas EQL, a neural-network-based symbolic regressor, tends to exhibit a more pronounced degradation in performance, often producing additional spurious terms and less accurate coefficient estimates. Overall, the proposed framework is applicable to systems with long-range interactions, provided that the patch size is large enough to capture the relevant dependencies.

---

> > ### Author Response · Authors · 2026-06-11
> >
> > In addition to the extra experiments described above, the reviewer thinks that it might also be necessary for the authors to revise on the manuscript's presentation. Specifically, the authors are encouraged to consider adding a subsection to describe the basics of symbolic regression with mathematical formulas. As far as the reviewer knows, this is not obvious to everyone in the general ML community. Therefore, adding such a short description might enhance the exposure of the manuscript.
> >
> > >We thank the reviewer for this helpful suggestion. We agree that symbolic regression may not be familiar to all readers in the broader machine learning community, and that a brief mathematical introduction can improve the accessibility of the manuscript.
> > >
> > >------
> > >In the revised manuscript, we have expanded the discussion under the **"Symbolic Regression"** subsection of the Framework section. Specifically, we now provide a concise mathematical formulation of the symbolic regression problem, including the objective, the associated optimization problem over the space of symbolic expressions, and the role of commonly used loss functions such as the mean squared error. We also clarify how symbolic regression differs from conventional regression by simultaneously searching for both the structure of the equation and its parameters.
> > >
> > >------
> > >We believe this additional background improves the exposition of the manuscript and makes the overall framework more accessible to readers who may not be familiar with symbolic regression techniques.

---

> > > ### Comment · Reviewer_hHfF · 2026-06-12
> > > **Response**
> > >
> > > The reviewer would like to thank the authors for the detailed rebuttal. The reviewer will make sure to examine all rebuttals and other reviews before submitting the final recommendation.

---

### Review · Reviewer_h98M · 2026-04-26

**Summary Of Contributions:**

This paper proposes a two-stage pipeline for estimating symbolic energy functions for unnormalized probability distributions from samples. In the first stage, the authors train a deep generative model (normalizing flow or score-based) on samples from the target distribution. In the second stage, they use the trained model to provide supervised targets for symbolic regression. The symbolic regressor is then used to obtain an analytic expression for the Hamiltonian or energy function.

The authors introduce a local factorization strategy for lattice systems: rather than learn a full high-dimensional Hamiltonian directly, they assume that the energy can be expressed as a sum of local patch-level terms. The empirical section applies the method to numerous examples inspired from physics.

**Audience:**

Yes

**Audience Explanation:**

Symbolic model discovery would certainly be of interest to the TMLR audience.

**Broader Impact Concerns:**

None.

**Claims And Evidence:**

No

**Claims Explanation:**

A central concern is the amount of prior structural information assumed. It seems that the patch structure and primitive function set are chosen to closely match the known target Hamiltonian, which makes the task much easier than general symbolic density discovery. The paper should be much more explicit and modest about the level of prior knowledge being used.

The quantitative metrics only measure functional approximation on the sampled region, not symbolic recovery. In the many-well and XY tables, some expressions have extra terms or distorted arguments but are still counted as successful based on predictive metrics. The paper needs a more rigorous distinction between “approximating the energy on test samples” and “recovering the true functional form”.

The renormalization section claims to discover new effective actions in a non-perturbative regime, but this is not supported. The current evidence is insufficient to establish discovery of a physically correct effective action.

Overall, the current manuscript overstates the generality and discovery aspect of the approach.

**Requested Changes:**

The current manuscript presents the whole pipeline as a new symbolic density estimator, but the algorithmic novelty relative to prior work is unclear. The authors should state precisely what is new: the use of symbolic regression on flow likelihoods? Local factorization of the objective? Etc. The paper does not compare against broader work on symbolic distillation from neural networks.

Make all inductive biases explicit. The paper should list, for every experiment: the operator library, the maximum expression complexity, the patch size, whether the patch size matches the true interaction range, whether the primitive set contains the exact true terms, etc. This is especially important because the method’s success depends heavily on the chosen primitives. Including power-four primitives for many-well and phi-four theory, product primitives for interaction terms, and trigonometric primitives for the XY model is a strong form of prior knowledge. The paper should present this as domain knowledge rather than as general symbolic discovery.

Add ablations that test whether the method can discover structure rather than only fit within a pre-specified structure. For example, use larger-than-necessary patches and quantify whether irrelevant long-range terms are suppressed. Include at least one benchmark whose true Hamiltonian is not exactly represented by the chosen primitive library, to distinguish approximation from exact symbolic recovery. These experiments would help address the concern that the method succeeds because the correct Hamiltonian form is already implicitly specified. Moreover, in the presence of misspecification, it should be tested whether the error propagation claims (that symbolic regression reduces the MSE relative to the flow model) still hold.

Evaluate symbolic recovery directly. Add metrics such as coefficient error for matched terms,
number of spurious terms, whether the recovered local interaction graph is correct, etc.

Other comments:

Abstract:
- “such as, factorizing large distributions” should be “such as factorizing large distributions”
- “computational Physics” should be “computational physics”
- “discovers new expressions for action function” should be “discovers new expressions for the action function”
- The claim that the new expressions are “intractable by traditional analytic approaches” is not validated

Introduction:
- “system identification problem” should be “the system identification problem”
- “upto 10” should be “up to 10”
- “flow-based and score-based model” should be plural: “models”
- “recover the energy functions, $H(x)$ We also derive...” is missing a period after $H(x)$
- “hamiltonians” should be capitalized
- “Again, deep generative models sample from high density regions...” should be  “Moreover” or “However”

Framework section:
- “Many a times” should be “Many times”
- “Model minimizes” should be “The model minimizes”
- “jacobian” should be “Jacobian”
- “due the high cost” should be “due to the high cost”
- “Unnormalised” and “Unnormalized” are used inconsistently
- Eq. (2) should have $\log \hat Z$, not $\hat Z$.
- Eq. (6): the expectation should be over both $\mathbf x$ and $\tilde{\mathbf x}$.
- Pg. 4: The paper says that the symbolic expression for the score can be integrated to obtain the Hamiltonian, but this is only true if the learned vector field is conservative.

Case studies:
- “Scalar $\phi^4$ Theory. It is a widely used...” should be “Scalar $\phi^4$ theory is widely used...”
- “Renomalization” is misspelled; it should be “Renormalization”
- The notation $N$ is used both for sample count and lattice size
- “to those lattice scale” should be “to those lattice scales”

Experiments:
- The manuscript alternates between “normalizing/normalising,” “modeling/modelling,” “maximizing/maximising,” “unnormalized/unnormalised”
- MSE values are reported as 0.00; more significant digits are needed
- Table 1: the target function for the second Gaussian in the caption does not match $\Sigma_2$.
- Table 3 has a malformed DSM + EQL expression with unmatched parentheses.
- Table 3 with $\lambda = 1/1.4$ does not match eq. (12).
- What does it mean forMeSSY to fail to converge?
- According to the analysis in Appendix D, the tables of symbolic expressions really ought to include variability across seeds.
- It seems that Appendix E does not match eq. (14).

Appendices:
- Appendix A.1: the covariance matrix is not diagonal.
- Appendix B.1 says the Gaussian base distribution has support over $\mathbf R^1$; is this meant to be one dimension?
- Appendix B: “flow modelling ,” has an extra space before the comma
- Table 7 should say sample size, not ensemble size

---

> ### Author Response · Authors · 2026-06-11
>
> The current manuscript presents the whole pipeline as a new symbolic density estimator, but the algorithmic novelty relative to prior work is unclear. The authors should state precisely what is new: the use of symbolic regression on flow likelihoods? Local factorization of the objective? Etc. The paper does not compare against broader work on symbolic distillation from neural networks.
>
>
> > We thank the reviewer for this important comment. In the revised manuscript, we have clarified both the scope of our claims and the specific methodological novelties of the proposed framework.
> >
> > First, we emphasize that our contribution is **not** a fundamentally new symbolic regression algorithm or a standalone symbolic density estimator. Rather, the novelty lies in combining modern score-/flow-based generative modeling with symbolic regression to recover interpretable energy or Hamiltonian function directly from the samples of the unnormalised probability distribution.
> >
> > More precisely, the main contributions are:
> >
> > 1. **Symbolic recovery of energies/Hamiltonians from unnormalized distributions using samples only.**
> > Traditional symbolic regression (SR) methods operate in a supervised setting where paired observations $(x,y)$ are available and the goal is to recover an analytical relationship $y=f(x)$. In contrast, our problem is fundamentally unsupervised: we are given only samples $x \sim p(x)$ from an unknown unnormalized distribution, $p(x)\propto e^{-H(x)}$  and seek to recover the underlying energy/Hamiltonian $H(x)$.> Existing SR methods cannot be directly applied in this setting because no explicit target variable $y$ is observed. To address this challenge, we propose a framework that transform the unsupervised density-learning problem into a supervised regression problem using generative models, specifically flow-based and score-based models. These models provide quantities such as log-likelihoods and score functions, that are directly related to the underlying energy function. Once these surrogate targets are obtained, symbolic regression can be applied to recover analytical expressions for the Hamiltonian. Thus, the key contribution is a framework that enables symbolic identification of unnormalized distributions using only samples from the distribution and quantities induced by learned generative models.
> >
> > 2. **Local factorization for scalable symbolic recovery in high-dimensional systems.**
> > Standard symbolic regression becomes computationally difficult and often produces uninterpretable expressions in high-dimensional settings ($d>10$).To address this, we propose factorizing the Hamiltonian into local Hamiltonians, assuming local dependencies and neighboring interactions. This approach allows us to learn the Hamiltonian of a high-dimensional distribution by learning low-dimensional local Hamiltonians instead of a single global expression, the proposed framework substantially reduces the complexity of symbolic recovery, improves interpretability, and enables tractable learning in structured many-body systems.
> >
> > 3. **Multi-scale perturbation for improved recovery of energy landscapes.**
> > Recovering the equation of a distribution from samples is challenging because empirical data is typically concentrated in high-density regions, providing limited information about the global structure of the energy landscape. To mitigate this issue, we introduce a multi-scale perturbation strategy in which Gaussian noise of varying magnitudes is added to the observed samples. This generates informative samples in lower-density regions and exposes the symbolic regression model to a broader portion of the distribution, leading to more stable and accurate recovery of the underlying Hamiltonian.
> >
> >
> > We would like to clarify that our work does not propose a new symbolic regression (SR) algorithm or a new symbolic distillation method at the optimization level. Instead, symbolic regression is used as a modular component within the overall framework, and in principle the proposed pipeline can be combined with a variety of existing SR methods.
> >
> >The primary contribution of the paper is therefore not the development of a new SR technique, but rather a framework for recovering interpretable energy functions and effective actions from generative models trained on samples from unnormalized distributions.
> >
> >In the revised manuscript, we therefore position our contribution more precisely as a framework for *symbolic interpretation/distillation of learned generative dynamics*, rather than as a completely new symbolic regression methodology in the introduction section.

---

> > ### Author Response · Authors · 2026-06-11
> >
> > Make all inductive biases explicit. The paper should list, for every experiment: the operator library, the maximum expression complexity, the patch size, whether the patch size matches the true interaction range, whether the primitive set contains the exact true terms, etc. This is especially important because the method’s success depends heavily on the chosen primitives. Including power-four primitives for many-well and phi-four theory, product primitives for interaction terms, and trigonometric primitives for the XY model is a strong form of prior knowledge. The paper should present this as domain knowledge rather than as general symbolic discovery.
> >
> > >We thank the reviewer for this important comment. We agree that the inductive biases introduced through the symbolic regression stage play a significant role in the success of the framework and should be stated explicitly. In particular, the choice of primitive operators, expression complexity, and patch size encode domain knowledge that constrains the hypothesis space explored by the symbolic regressor.
> > >
> > >-----
> > >Several of these implementation details were already provided in the experimental sections of the main paper and in Appendices B.3 and B.4. However, to improve clarity and transparency, we have consolidated this information into a dedicated summary table in the revised manuscript. The table explicitly reports, for each benchmark, the operator library, model complexity, patch size, and whether the selected patch size matches the true interaction range.
> > >
> > >-------
> > >For EQL, the symbolic regressor is implemented as a fully connected neural network in which conventional activation functions are replaced by symbolic building blocks (e.g., $\sin(\cdot)$, $\cos(\cdot)$, identity, multiplication, polynomial operators). The effective expression complexity is controlled through the network depth (no. of layers) and the number of neurons assigned to each symbolic operator. The quantities shown in parentheses in the table denote the number of neurons corresponding to each operator.
> > >
> > >-------
> > >| Experiment            | Operator Library                                                          | No of layers | Patch Size  | Range of Interaction |
> > >|-----------------------|---------------------------------------------------------------------------|--------------|-------------|----------------------|
> > >| Multivariate Gaussian | Constant (2), Identity (2), Cos (1), Square (2), exp(2), * (2), pow-4 (2) | 2/3          | None        | -                    |
> > >| Many Well             | Constant (2), Identity (2), Square (3), exp (2), * (1), pow-4 (2)         | 2/3          | 2x1         | 2x1                  |
> > >| Phi-4                 | Constant (1), Identity (2), Square (2), exp (1), * (2), pow-4 (2)         | 2            | 2x2,3x3,4x4 | 2x2                  |
> > >| XY                    | Constant (1), Identity (1), Cos (1), Sin (1), Square (1), * (2)           | 2            | 2x2,3x3     | 2x2                  |
> > >
> > >
> > >
> > > For PySR, a genetic-programming-based symbolic regression algorithm, the inductive bias is specified through the choice of primitive operators, the maximum expression complexity, and the population size used during the evolutionary search. These choices determine the hypothesis space explored by the symbolic regressor and therefore directly influence the expressions that can be recovered. We summarize the PySR configuration used for each benchmark in the table below.
> > >
> > >
> > >| Experiment            | Operator Library   | Population Size | Complexity | Patch Size  | Range of Interaction |
> > >|-----------------------|--------------------|-----------------|------------|-------------|----------------------|
> > >| Multivariate Gaussian | {+,-,*}            | 10/20             | 20         | None        | -                    |
> > >| Many Well             | {+,-,*, pow-2}     | 30              | 30         | 2x1         | 2x1                  |
> > >| Phi-4                 | {+,-,*, pow-2,3,4} | 30              | 30         | 2x2,3x3,4x4 | 2x2                  |
> > >| XY                    | {+,-,*,Sin, Cos}   | 30              | 30         | 2x2,3x3     | 2x2                  |
> > >
> > >
> > > We would also like to emphasize that these operator choices were not intended to represent unconstrained symbolic discovery. Rather, they encode physically motivated prior knowledge about the class of systems under consideration. For example, quartic operators are included for the $\phi^4$ and many-well systems because quartic energy terms are known to be present, product operators are required to represent interaction terms, and trigonometric operators are included for the XY model because its Hamiltonian is naturally expressed through cosine interactions. Accordingly, in the revised manuscript we have revised the presentation to explicitly characterize these choices as domain-informed inductive biases rather than assumptions-free symbolic discovery.

---

> > > ### Author Response · Authors · 2026-06-11
> > >
> > > Add ablations that test whether the method can discover structure rather than only fit within a pre-specified structure...
> > > > We have done an experiment, where we use larger than necessary patches on scalar $\phi^4$ dataset. We assess the evaluation with larger patches, i.e., $3\times3$ and $4\times4$ to test the generalisability of the method. The results for $\lambda_1 = 4, \lambda_2 = 1$ with the flow model using different patch sizes are reported in Table 5. Performance remains strong for both PySR and EQL, though larger patches lead to increased search complexity. These experiments support that the SR methods are able to recover the true equation, or at least the important interaction terms thereof, even with larger patches
> > > >
> > > >|      | $\|p\|$    | MSE | $R^2$| Estimated $\hat h(x_{p(l)})$                                                                                                                                                                                                                                   |
> > > >|------|------------|-------------------|-----------------|---------------------------------------------------------------------------------------------------------------------------------------------------------------------------------------------------------------------------------------------------------------|
> > > >| PySR | $2\times2$ | 0.003             | 0.99            | $ 4.01x_{0}^{4} + 5.17x_{0}^2 - 2.06x_{0}x_{1} - 2.06x_{0}x_{2}$                                                                                                                                                                                              |
> > > >|      | $3\times3$ | 0.006             | 0.99            | $4.11x_{0}^4 + 5.06x_{0}^2 - 2.04x_{0}x_{1} - 2.04x_{0}x_{2} $                                                                                                                                                                                                |
> > > >|      | $4\times4$ | 0.008             | 0.99            | $3.90 x_0^4 + 5.14 x_0^2 -1.99 x_0x_1 -1.99x_0x_2$                                                                                                                                                                                                            |
> > > >| EQL  | $2\times2$ | 0.07              | 0.99            | $3.69 x_0^4 +5.27 x_0^2-2.01 x_0 x_2 -2.14 x_0 x_1 -\textcolor{gray}{0.68 x_1 x_2^3-0.27 x_2^3 x_3+0.14 x_0 x_3+0.45 x_1 x_2}$                                                                       |
> > > >|      | $3\times3$ | 0.98              | 0.98            | $3.86x_0^4 +5.21x_0^2-2.1x_0x_1-1.99x_0x_2 + \textcolor{gray}{0.42x_1x_2-1.01x_0^3x_2-0.14x_0^3x_1-0.13x_0^3x_5}$                                                                                      |
> > > >|      | $4\times4$ | 0.11              | 0.99            | $3.61x_0^4 +4.97x_0^2-2.09x_0x_1 -2.03x_0x_2 +\textcolor{gray}{0.11x_0^3x_3} -\textcolor{gray}{0.12x_0^3x_8}-\textcolor{gray}{0.1x_0x_9+0.28x_1x_4+0.27x_0x_5+0.14x_0x_2+0.13x_0x_8}$ |
> > > >
> > > >These results provide additional evidence that the framework is capable of identifying the relevant interaction structure even when the patch size is larger than necessary. In particular, PySR consistently recovers the exact local Hamiltonian across all patch sizes without introducing any irrelevant interaction terms, despite the substantially larger search space associated with the $3\times3$ and $4\times4$ patches. This indicates that the method is not merely fitting within a pre-specified structure, but is able to suppress unnecessary long-range interactions and identify the physically relevant terms.
> > > >
> > > >------
> > > >In contrast, EQL recovers the dominant Hamiltonian terms in all cases but introduces an increasing number of spurious interactions as the patch size grows. Notably, many of these additional terms involve variables located farther from the central site, corresponding to interactions that are absent in the true nearest-neighbor Hamiltonian. This trend suggests that, as the dimensions of the symbolic search space increases, EQL becomes more prone to fitting irrelevant long-range dependencies. Nevertheless, the coefficients of the true interaction terms remain close to their ground-truth values, indicating that the physically meaningful structure is still largely recovered.
> > > >
> > > >------
> > > >Overall, these ablation results demonstrate that the proposed framework can distinguish relevant from irrelevant interactions when provided with a larger-than-necessary local neighborhood. The ability of PySR to recover the correct Hamiltonian without introducing additional long-range terms provides direct evidence that the method is capable of discovering interaction structure rather than merely fitting expressions within a prescribed patch.

---

> > > > ### Author Response · Authors · 2026-06-11
> > > >
> > > > Include at least one benchmark whose true Hamiltonian is not exactly represented by the chosen primitive library, to distinguish approximation from exact symbolic recovery. These experiments would help address the concern that the method succeeds because the correct Hamiltonian form is already implicitly specified.
> > > >
> > > > > We thank the reviewer for this insightful suggestion. We agree that it is important to distinguish between **exact symbolic recovery** and **approximate symbolic distillation** when the true Hamiltonian is not contained within the chosen primitive library.
> > > > >
> > > > >------
> > > > >To address this concern, we conducted an additional experiment on the scalar $\phi^4$ benchmark using a restricted symbolic library. Specifically, we removed the explicit polynomial primitives $\{x, x^2, x^3, x^4\}$ and instead provided only the standard binary operators $\{+, -, \times, /\}$ together with the unary operators $\{\sin, \cos\}$.
> > > > >
> > > > >
> > > > >|           | MSE | $R^2$ | Estimated $\hat{h}(x_{p(l)})$                                                                                                                                 |
> > > > >|-----------|-------------------|-------------------|---------------------------------------------------------------------------------------------------------------------------------------------------------------|
> > > > >| NF + PySR | 0.0085            | 0.99              | $3.89x_{0}^4 + 5.15x_{0}^2 -2.00x_{0}x_{1} -2.00x_{0}x_{2}$                                                                                                   |
> > > > >| NF + EQL  | 2.2872            | 0.90              | $5.10x_{0}^2 -2.16x_{0}x_{1} -1.75x_{0}x_{2} +2.62x_{0}x_{3} +0.25x_{1}x_{2}+\cdots+\text{higher order polynomial terms}+\cdots+\text{Sin, Cos nested terms}$ |
> > > > >
> > > > >------
> > > > >The results show that the framework can still recover a meaningful approximation of the underlying Hamiltonian even when the exact functional form is not explicitly available in the primitive library. PySR remains able to construct an accurate approximation through compositions of the available operators, achieving a high $R^2$ score and recovering the dominant interaction structure. In contrast, EQL exhibits a noticeable degradation in performance and introduces additional spurious terms, indicating that symbolic recovery becomes more challenging when the hypothesis space is mismatched to the true Hamiltonian.
> > > > >
> > > > >------
> > > > >These experiments highlight that the success of the framework depends not only on the quality of the learned generative model but also on the expressivity of the symbolic library. When the true Hamiltonian is not representable within the available primitives, the recovered expression should be interpreted as an **effective symbolic approximation** rather than an exact recovery of the underlying physics.
> > > > >
> > > > >We have added this experiment and corresponding discussion to the revised manuscript in the appendix and have clarified the role of inductive biases introduced through the choice of primitive library.
> > > >
> > > >
> > > > Moreover, in the presence of misspecification, it should be tested whether the error propagation claims (that symbolic regression reduces the MSE relative to the flow model) still hold.
> > > >
> > > > >We compared the error of the learned flow model, $\mathrm{MSE}(H,H_\theta)$, against the error of the symbolic approximation, $\mathrm{MSE}(H,\hat H)$, when the true Hamiltonian is not exactly representable by the chosen primitive library. The table below shows the error propagation analysis for the above experiment.
> > > > >| Dataset   | MSE$(H,H_\theta)$ | MSE$(H,\hat H)$ |
> > > > >|-----------|:-----------------:|:---------------:|
> > > > >| NF + PySR |        0.22       |       0.01      |
> > > > >| NF + EQL  |        0.22       |       2.29      |
> > > > >
> > > > >------
> > > > >The results show that the error-reduction property is not automatic and depends on the ability of the symbolic regressor to construct an effective approximation within the available hypothesis space. In the case of PySR, symbolic regression substantially reduces the error relative to the underlying flow model, despite the primitive library being misspecified. This suggests that PySR is able to identify a compact symbolic approximation that captures the dominant structure of the learned Hamiltonian. In contrast, EQL introduces several additional misspecified terms, resulting in a significantly larger error than that of the original flow model. Thus, under primitive set misspecification, symbolic regression can either reduce or amplify errors depending on the expressiveness and robustness of the underlying SR algorithm.
> > > > >
> > > > >------
> > > > >We have added this analysis to the revised manuscript and clarified that the error-propagation claims hold when the symbolic regressor is able to construct an accurate approximation of the learned generative model, but are not guaranteed under severe primitive set mismatch.

---

> > > > > ### Author Response · Authors · 2026-06-11
> > > > >
> > > > > Evaluate symbolic recovery directly. Add metrics such as coefficient error for matched terms, number of spurious terms, whether the recovered local interaction graph is correct, etc.
> > > > >
> > > > > >We thank the reviewer for this valuable suggestion. We agree that reconstruction metrics such as MSE and $R^2$ alone do not fully characterize the quality of symbolic recovery. Since the primary objective of our framework is to recover interpretable symbolic representations of the underlying Hamiltonian, we have added several additional metrics that directly evaluate the recovered expressions against the ground-truth symbolic form.
> > > > > >
> > > > > >------
> > > > > >Specifically, the revised manuscript now reports:
> > > > > >
> > > > > >- **Relative Coefficient Error (RCE):** Measures the accuracy of the recovered coefficients for correctly matched symbolic terms. Let $\mathcal{M}$ denote the set of matched terms, $c_i$ the ground-truth coefficient of the $i$-th term, and $\hat{c}_i$  the corresponding recovered coefficient.
> > > > > >
> > > > > >$$ \mathrm{RCE}=\frac{1}{|\mathcal{M}|}\sum_{i\in\mathcal{M}}\frac{|\hat c_i-c_i|}{|c_i|}$$
> > > > > >where $\mathcal{M}$ denotes the set of matched terms, $c_i$ is the ground-truth coefficient, and $\hat{c}_i$ is the recovered coefficient.
> > > > > >An RCE value of zero corresponds to exact coefficient recovery, while larger values indicate greater deviation of the recovered coefficients from the ground truth.
> > > > > >
> > > > > >- **Precision:** Measures the fraction of recovered terms that correspond to true terms,
> > > > > >$$\text{Precision} = \frac{\mathrm{TP}}{\mathrm{TP}+\mathrm{FP}}$$
> > > > > >where TP denotes the number of correctly recovered terms and FP denotes the number of spurious terms.
> > > > > >
> > > > > >- **Recall:** Measures the fraction of true terms that are successfully recovered,
> > > > > >$$\text{Recall} = \frac{\mathrm{TP}}{\mathrm{TP}+\mathrm{FN}}$$
> > > > > >where FN denotes the number of ground-truth terms that were not recovered.
> > > > > >
> > > > > >- **F1 Score:** Harmonic mean of precision and recall,
> > > > > >$$\text{F1} = \frac{2\,(\text{Precision})(\text{Recall})}{\text{Precision}+\text{Recall}}$$
> > > > > >
> > > > > >- **Number of Spurious Terms:** The total number of symbolic terms present in the recovered expression that do not appear in the ground-truth Hamiltonian.
> > > > > >
> > > > > >------
> > > > > >These metrics provide a more direct assessment of symbolic recovery quality by quantifying not only predictive accuracy but also the correctness of the recovered symbolic structure. In particular, they help distinguish cases where a model achieves good MSE/$R^2$ values but introduces incorrect interactions or inaccurate coefficients.
> > > > > >
> > > > > >------
> > > > > >Following the reviewer's suggestion, we have added a dedicated appendix section defining these metrics and have included supplementary tables reporting them for the benchmark problems considered in the paper.

---

> > > > > > ### Author Response · Authors · 2026-06-11
> > > > > >
> > > > > > The renormalization section claims to discover new effective actions in a non-perturbative regime, but this is not supported. The current evidence is insufficient to establish discovery of a physically correct effective action.
> > > > > >
> > > > > > >To strengthen the validation, we have added quantitative physical-observable based evaluations in the revised manuscript and have also softened the corresponding claims throughout the paper.
> > > > > > >
> > > > > > >------
> > > > > > >Specifically, for the RG $\phi^4$ experiments, we use the recovered Hamiltonians to generate new samples via HMC and compare the resulting distributions against the original coarse-grained data. We evaluate following observables, that include magnetization $M$, absolute magnetization $|M|$, and magnetic susceptibility $\chi$, for lattice sizes $32\times32$, $16\times16$, and $8\times8$.
> > > > > > >
> > > > > > >------
> > > > > > >To quantify agreement between the coarse-grained data and samples generated from the recovered Hamiltonians, we compute (i) the Earth Mover's Distance (EMD) between the observable distributions and (ii) the percentage overlap between the corresponding histograms. The results show consistently high agreement across all observables and lattice sizes. The histogram overlap exceeds $98\%$ in every case and reaches more than $99\%$ for magnetic susceptibility at the smaller lattice sizes. Correspondingly, the EMD values remain very small, ranging from approximately $10^{-4}$ to $10^{-3}$, indicating that the generated distributions closely match the coarse-grained distributions.
> > > > > > >
> > > > > > >|                               | 32x32                 |                    | 16x16                 |                    | 8x8                   |                    |
> > > > > > >|-------------------------------|-----------------------|--------------------|-----------------------|--------------------|-----------------------|--------------------|
> > > > > > >|           Observable          | % Overlap($\uparrow$) | EMD ($\downarrow$) | % Overlap($\uparrow$) | EMD ($\downarrow$) | % Overlap($\uparrow$) | EMD ($\downarrow$) |
> > > > > > >| Magnetization, M              | 98.116                | 2.28 X 10^{-4}     | 98.845                | 9.09 X 10^{-5}     | 98.759                | 8.79 X 10^{-5}     |
> > > > > > >| Absolute Magnetization, \|M\| | 98.197                | 2.06 X 10^{-4}     | 98.794                | 8.69 X 10^{-5}     | 98.749                | 8.30 X 10^{-5}     |
> > > > > > >| Magnetic Susceptibility       | 98.244                | 3.81 X 10^{-3}     | 99.187                | 4.47 X 10^{-4}     | 99.187                | 1.13 X 10^{-4}     |
> > > > > >
> > > > > >
> > > > > > >|              |                |   Magnetization, M   |                    | Absolute Magnetization, \|M\| |                    | Magnetic Susceptibility |                    |
> > > > > > >|:------------:|----------------|:--------------------:|:------------------:|:-----------------------------:|:------------------:|:-----------------------:|:------------------:|
> > > > > > >|              |                |         Mean         | Std Dev  |              Mean             | Std Dev |           Mean          | Std Dev  |
> > > > > > >| $32\times32$ | Coarsened Data | $-1.82\times10^{-6}$ |       0.0066       |             0.0053            |       0.0039       |          0.0441         |       0.0609       |
> > > > > > >|              | Generated Data |  $9.09\times10^{-5}$ |       0.0068       |             0.0055            |       0.0041       |          0.0481         |       0.0681       |
> > > > > > >| $16\times16$ | Coarsened Data | $-1.82\times10^{-6}$ |       0.0066       |             0.0053            |       0.0039       |          0.0110         |       0.0152       |
> > > > > > >|              | Generated Data | $-3.36\times10^{-5}$ |       0.0067       |             0.0053            |       0.0040       |          0.0115         |       0.0161       |
> > > > > > >|  $8\times8$  | Coarsened Data | $-1.82\times10^{-6}$ |       0.0066       |             0.0053            |       0.0039       |          0.0028         |       0.0038       |
> > > > > > >|              | Generated Data | $-2.31\times10^{-5}$ |       0.0067       |             0.0053            |       0.0040       |          0.0029         |       0.0041       |
> > > > > > >
> > > > > > >In addition, we compare the first and second moments of these observables. For all lattice sizes considered, the means and standard deviations computed from samples generated using the recovered Hamiltonians closely match those of the coarse-grained data.
> > > > > > >
> > > > > > >These results provide additional evidence that the recovered Hamiltonians capture not only the symbolic structure of the coarse-grained system but also reproduce key physical observables and their distributions. Accordingly, we have revised the manuscript to present the results more cautiously. Rather than claiming that the method definitively discovers new effective actions, we now state that the recovered symbolic Hamiltonians provide interpretable approximations of the effective coarse-grained action and are validated through agreement with observable statistics and sampling-based evaluations.
> > > > > > >
> > > > > > >

---

> > > > > > > ### Author Response · Authors · 2026-06-11
> > > > > > >
> > > > > > > ## Other comments:
> > > > > > >
> > > > > > > ### Abstract:
> > > > > > >
> > > > > > > - “such as, factorizing large distributions” should be “such as factorizing large distributions”
> > > > > > > > We have corrected this in the revised manuscript.
> > > > > > >
> > > > > > > - “computational Physics” should be “computational physics”
> > > > > > > > We have corrected this in the revised manuscript.
> > > > > > >
> > > > > > > - “discovers new expressions for action function” should be “discovers new expressions for the action function”
> > > > > > > > We have corrected this in the revised manuscript.
> > > > > > >
> > > > > > > - The claim that the new expressions are “intractable by traditional analytic approaches” is not validated
> > > > > > > > We thank the reviewer for pointing this out. We agree that the original wording was too strong and not sufficiently validated by the current experiments. Our intention was not to claim that the proposed method solves analytically intractable renormalization problems or discovers fundamentally new physical laws. Rather, we aim to demonstrate that the framework can recover compact symbolic approximations of effective action terms directly from samples in settings where deriving closed-form expressions can be challenging using standard analytic procedures.
> > > > > > >
> > > > > > > > Accordingly, we have softened the claim throughout the revised manuscript and clarified that the recovered expressions should be interpreted as data-driven symbolic approximations rather than rigorously established new analytic solutions.
> > > > > > >
> > > > > > > ### Introduction:
> > > > > > >
> > > > > > > - “system identification problem” should be “the system identification problem”
> > > > > > > > We have corrected this in the revised manuscript.
> > > > > > >
> > > > > > > - “upto 10” should be “up to 10”
> > > > > > > > We have corrected this in the revised manuscript.
> > > > > > >
> > > > > > > - “flow-based and score-based model” should be plural: “models”
> > > > > > > > We have corrected this in the revised manuscript.
> > > > > > >
> > > > > > > - “recover the energy functions, $H(x)$  We also derive...” is missing a period after $H(x)$
> > > > > > > > We have corrected this in the revised manuscript.
> > > > > > >
> > > > > > > - “hamiltonians” should be capitalized
> > > > > > > > We have corrected this in the revised manuscript.
> > > > > > >
> > > > > > > - “Again, deep generative models sample from high density regions...” should be “Moreover” or “However”
> > > > > > > > We have corrected this in the revised manuscript.
> > > > > > >
> > > > > > > ### Framework section:
> > > > > > >
> > > > > > > - “Many a times” should be “Many times”
> > > > > > > > We have corrected this in the revised manuscript.
> > > > > > >
> > > > > > > - “Model minimizes” should be “The model minimizes”
> > > > > > > > We have corrected this in the revised manuscript.
> > > > > > >
> > > > > > > - “jacobian” should be “Jacobian”
> > > > > > > > We have corrected this in the revised manuscript.
> > > > > > >
> > > > > > > - “due the high cost” should be “due to the high cost”
> > > > > > > > We have corrected this in the revised manuscript.
> > > > > > >
> > > > > > > - “Unnormalised” and “Unnormalized” are used inconsistently
> > > > > > > > We have corrected this in the revised manuscript. We are using Unnormalized now everywhere.
> > > > > > >
> > > > > > > - Eq. (2) should have $\log \hat{Z}$, not $\hat{Z}$.
> > > > > > > > We have corrected this in the revised manuscript.
> > > > > > >
> > > > > > > - Eq. (6): the expectation should be over both $x$ and $\tilde{x}$.
> > > > > > > > We have corrected this in the revised manuscript.
> > > > > > >
> > > > > > > - Pg. 4: The paper says that the symbolic expression for the score can be integrated to obtain the Hamiltonian, but this is only true if the learned vector field is conservative.
> > > > > > > > We thank the reviewer for pointing this out. We agree that the statement is strictly valid only when the learned score field is conservative (i.e., curl-free), since only then does there exist a scalar potential whose gradient corresponds to the recovered vector field. In our experiments, the score model is trained to approximate the gradient of the log-density, which is theoretically conservative for the target distributions considered. However, due to approximation and optimization errors, the learned score field may not be exactly conservative in practice. We have revised the manuscript to clarify this assumption and its practical limitations.
> > > > > > > ### Case studies:
> > > > > > >
> > > > > > > -“Scalar $\phi^4$ Theory. It is a widely used...” should be “Scalar $\phi^4$  theory is widely used...”
> > > > > > > > We have corrected this in the revised manuscript.
> > > > > > >
> > > > > > > -“Renomalization” is misspelled; it should be “Renormalization”
> > > > > > > > We have corrected this in the revised manuscript.
> > > > > > >
> > > > > > > -The notation $N$ is used both for sample count and lattice size
> > > > > > > > We have corrected this in the revised manuscript and used $N$ for sample count and $d$ for data dimensionality as well as lattice size.
> > > > > > >
> > > > > > > -“to those lattice scale” should be “to those lattice scales”
> > > > > > > > We have corrected this in the revised manuscript.

---

> ### Author Response · Authors · 2026-06-11
>
> ### Experiments:
>
> - The manuscript alternates between “normalizing/normalising,” “modeling/modelling,” “maximizing/maximising,” “unnormalized/unnormalised”
> > We have corrected this in the revised manuscript.
>
> - MSE values are reported as 0.00; more significant digits are needed.
> > We have revised the manuscript and updated Table 1 to report the MSE values with higher numerical precision and additional significant digits for improved clarity and comparison.
>
> - Table 1: the target function for the second Gaussian in the caption does not match $\Sigma_2$.
> > We thank the reviewer for carefully checking this point. We verified the expression in Table 1 and confirm that the target function is consistent with the covariance matrix $\Sigma_2$. In particular, the reported target Hamiltonian corresponds to the quadratic form
> $0.5\,\mathbf{X}^T\Sigma^{-1}\mathbf{X}$.
>
> - Table 3 has a malformed DSM + EQL expression with unmatched parentheses.
> > We have corrected this in the revised manuscript.
>
> - Table 3 with $\lambda = 1/1.4$  does not match eq. (12).
> > We thank the reviewer for carefully checking this point. We verified the consistency between Table 3 and Eq. (12). The apparent discrepancy arises because each interaction term appears twice in the summation of Eq. (12). After accounting for this double counting, the effective coefficient associated with each interaction term becomes $\lambda = 1/1.4 \approx 0.71$, which matches the expression reported in Table 3.
>
> - What does it mean for MeSSY to fail to converge?
> > By “failure to converge,” we mean that the optimization procedure of MeSSY does not terminate successfully within a reasonable computational budget. In our experiments, the algorithm repeatedly continued its search/optimization loop without producing a stable symbolic expression, even when trained using a very small number of samples for datasets with dimensions greater than two ($d>2$). In practice, this resulted in extremely long runtimes without convergence to a final model. We used the implementation provided by the authors in their official repository to run these experiments.
>
> - According to the analysis in Appendix D, the tables of symbolic expressions really ought to include variability across seeds.
> > In the revised manuscript, we have added the table including variability across seeeds in the Appendix G under robustness analysis.
>
> - It seems that Appendix E does not match eq. (14).
> > We thank the reviewer for pointing this out. We verified that the evaluation procedure in Appendix E is consistent with Eq. (14), although the presentation may have caused confusion. The key point is that the Hamiltonian recovered from the NF based framework as well as score-based formulation is defined only up to an additive constant. Besides integrating the predicted score field introduces an arbitrary integration constant. This constant shift does not affect the underlying energy landscape or probability distribution, but it can artificially influence the MSE and $R^2$ metrics if not removed prior to evaluation.
> >
> >Therefore, before computing the metrics, we subtract the empirical mean from both the target Hamiltonian and the recovered Hamiltonian:
> >$\tilde H(x)=H(x)-\bar H,$
> >$\tilde{\hat H}(x)=\hat H(x)-\bar{\hat H}.$
>
> >The MSE and $R^2$ metrics are then computed using these mean-normalized quantities. Thus, Appendix E does not modify Eq. (14), but rather makes explicit the preprocessing step used to remove additive constants prior to evaluation. We have revised the appendix E to clarify this point more explicitly and avoid ambiguity.
>
>
> ### Appendices:
>
> - Appendix A.1: the covariance matrix is not diagonal.
> > We have corrected this in the revised manuscript.
>
> - Appendix B.1 says the Gaussian base distribution has support over $\mathbb{R}^1$; is this meant to be one dimension?
> > Yes, the base Gaussian distribution in this specific setting is one-dimensional. To avoid ambiguity, we have replaced $\mathbb{R}^1$ with the more standard notation $\mathbb{R}$ in the revised manuscript.
>
> - Appendix B: “flow modelling ,” has an extra space before the comma
> > We have corrected this in the revised manuscript.
>
> - Table 7 should say sample size, not ensemble size
> > We have corrected this in the revised manuscript.

---

> > ### Comment · Reviewer_h98M · 2026-06-14
> >
> > Thank you for the response. The new revision substantially improves transparency and evaluation. However, some central limitations remain: strong performance still relies on significant domain knowledge to specify the libraries and patch structure; the baseline comparisons are limited; and the misspecification experiment seems strange to me, since even with polynomial terms removed, they can still be constructed through multiplication (so it is not truly misspecified).
> >
> > Overall, I would suggest moving the symbolic recovery metrics to the main text and to soften claims such as the discovery of "new actions", which cannot be asserted based on approximation error alone.

---

> ### Author Response · Authors · 2026-06-23
>
> > We thank the reviewer for the constructive feedback and are pleased that the revised manuscript improved transparency and evaluation.
> >
> >-------
> > **Dependence on Domain Knowledge**
> >
> > We agree that the current framework relies on domain knowledge in specifying the primitive library and patch structure. Our goal is not fully automated scientific discovery, but rather symbolic recovery guided by physically motivated inductive biases. We have clarified this limitation in the discussion section and now explicitly state that recovery quality depends on the expressiveness of the chosen primitive library.
> >
> >--------
> > **Misspecification Experiment**
> >
> > We thank the reviewer for pointing out this issue. We agree that our original experiment should not be regarded as a strictly misspecified setting. Although explicit polynomial terms (e.g., $x^2$,$x^4$) were removed from the library, they can still be reconstructed through repeated multiplication, meaning that the target Hamiltonian remains representable.
> >
> >------
> > We have revised the discussion accordingly and now describe the experiment as a test of the method's robustness when important basis functions are removed, forcing the symbolic regressor to reconstruct them indirectly, rather than as a true misspecification experiment. More broadly, misspecification in symbolic regression is often not absolute, since many functions can be reconstructed or approximated using compositions of other operators (e.g., polynomials via multiplication and trigonometric functions via polynomial approximations of taylor expansion); the key distinction is often the complexity of the resulting representation rather than strict representability.
> >
> >--------
> > To address the concern more directly, we additionally performed a stronger misspecification experiment on the XY model. The ground-truth Hamiltonian contains cosine interactions, but all trigonometric operators ($\sin,\cos$) were removed from the primitive library. In this setting, the symbolic regressor can only approximate the interaction indirectly through arithmetic operations and polynomial compositions. The results in the Table below indicate that the framework can still recover meaningful approximations despite the absence of the true functional form. These new results have been added to the manuscript.
> >
> >The actual target hamiltonian expression for XY model dataset is
>
> $$
> h(x_{p(l)}) = -0.71\cos(x_0-x_1) - 0.71\cos(x_0-x_2)
> $$
>
> $$
> \approx -0.71\left(1-\frac{(x_0-x_1)^2}{2!}+\frac{(x_0-x_1)^4}{4!}+\cdots\right)
> -0.71\left(1-\frac{(x_0-x_2)^2}{2!}+\frac{(x_0-x_2)^4}{4!}+\cdots\right)
> $$
>
> $$
> \approx -1.42 + 0.71x_0^2 +0.36x_1^2+0.36x_2^2-0.71x_0x_1-0.71x_0x_2+0.06x_0^4+0.03x_1^4+0.03x_2^4-0.12x_0^3x_1-0.12x_0^3x_2+0.12x_0x_1^3-0.12x_0x_2^3+0.18x_0^2x_1^2+0.18x_0^2x_2^2+\cdots
> $$
> >
> >
> >|         | MSE(\downarrow) | $R^2(\uparrow)$ | Estimated  $\hat{h}(x_{p(l)})$                                                                                                                                     |
> >|---------|-----------------|-----------------|--------------------------------------------------------------------------------------------------------------------------------------------------------------------|
> >| NF+PySR | 8.84            | 0.8088          | $0.15x_0^2+0.56x_1^2-0.43x_0x_1-0.30x_0x_2-0.07x_1^3-0.07x_0x_1^2+0.15x_0^2x_1-0.03x_0^2x_1^2-0.03x_0^2x_2^2+0.03x_0x_1^3+0.02x_0x_2^3+0.02x_0^3x_2 +0.15x_1+0.88$ |
> >| NF+EQL  | 7.13            | 0.8463          | $0.46x_0^2+0.14x_1^2+0.17x_2^2-0.37x_0x_1-0.24x_0x_2+0.14x_1+0.11x_0+0.59$                                                                                         |
> >
> >The quantitative results in above table show that both symbolic regression methods are able to recover meaningful approximations of the underlying cosine interaction despite the absence of trigonometric primitives in the primitive library. Although the recovered expressions do not match the true Hamiltonian exactly, they capture several dominant low-order terms expected from the Taylor expansion of the cosine function, including quadratic self-interaction terms and pairwise interaction terms such as $x_0x_1$ and $x_0x_2$. The resulting approximations achieve reasonably high coefficients of determination ($R^2=0.81$ for PySR and $R^2=0.85$ for EQL), indicating that a substantial fraction of the Hamiltonian variance is explained even under strong library misspecification. Interestingly, EQL attains a slightly higher $R^2$ and lower MSE than PySR in this setting, suggesting that simpler low-order polynomial approximations may provide a more robust surrogate when the true trigonometric functional form is unavailable. These results further support the view that symbolic recovery quality depends not only on exact representability but also on the ability of the symbolic regressor to construct effective approximations within the restricted hypothesis space.

---

> ### Author Response · Authors · 2026-06-23
>
> > **Claims Regarding “New Actions”**
> >
> > We agree that approximation accuracy alone is insufficient to support claims of discovering genuinely new effective actions. Following the reviewer's suggestion, we have softened the corresponding statements throughout the manuscript. The revised manuscript now frames the recovered expressions as interpretable symbolic approximations of effective actions or Hamiltonians rather than exact representations of the underlying physical system.
> >
> >-------
> >For the **renormalization group experiments in $\phi^4$ theory**, we  further assess the physical relevance of the recovered Hamiltonians by generating samples via Hamiltonian Monte Carlo (HMC) and compare the physical observables such as magnetization, absolute magnetization, and susceptibility, with those obtained from the original coarse-grained data across multiple lattice sizes. **The resulting Earth Mover's Distance (EMD) and histogram overlap metrics indicate strong agreement between the two distributions**. These results provide complementary evidence that the recovered Hamiltonians reproduce important observable statistics of the coarse-grained system. Further details are presented in Tables 6 and 7 of the revised manuscript.
> >
> >----------
> > **Following the reviewer's suggestion, symbolic recovery metrics have also been moved from the appendix to the main text.**
> >
> >----------
> > **Limited Baselines**
> >
> >We agree that the number of available baselines is limited. To the best of our knowledge, there are currently very few methods specifically designed for symbolic density estimation, and no widely adopted alternatives that are directly applicable to the settings considered in this work. We therefore evaluated the closest available baseline, MeSSY and included the corresponding results in the manuscript.
> >
> >
> >---------
> >We also investigated scaling MeSSY to higher-dimensional problems using the authors' official implementation. However, the method relies on numerical integration and  polynomial basis functions whose size grows rapidly with dimensionality, leading to a substantial increase in computational cost as the problem dimension increases. In our experiments, this prevented successful application beyond very low-dimensional settings within a reasonable computational budget. We have clarified this limitation in the revised manuscript and now explicitly discuss the restricted baseline landscape as a limitation of the current study.
> >
> >---------
> > We thank the reviewer for these suggestions, which have improved both the clarity and precision of the manuscript.

---

### Review · Reviewer_514Y · 2026-05-18

**Summary Of Contributions:**

This paper proposes a general framework to integrate deep generative modeling, either a likelihood-based normalizing flow or a score-based model, with symbolic regression to estimate probability density functions as symbolic expression from samples. The framework also introduces useful inductive biases, including local factorization over lattice patches and noisy perturbation of samples to expose the symbolic regression stage to low-density/high-energy regions. The most interesting contribution is the combination of deep generative density or score estimation with symbolic regression for interpretable unnormalized density estimation, together with a physics-motivated factorization strategy that makes high-dimensional lattice problems tractable.

**Audience:**

Yes

**Audience Explanation:**

At least some TMLR readers would likely be interested in this work. The paper sits at the intersection of interpretable machine learning, deep generative modeling, symbolic regression, and scientific discovery. The problem of converting black-box generative models into explicit symbolic density or energy functions is relevant to researchers working on scientific ML, probabilistic modeling, energy-based models, and physics-informed learning. The use of normalizing flows and score models as intermediate estimators is also natural and timely, and the factorized symbolic regression strategy is a practical idea for scaling symbolic discovery to structured high-dimensional systems.

**Claims And Evidence:**

Yes

**Claims Explanation:**

The submission provides reasonably clear evidence for the narrower claim that the proposed two-stage pipeline can recover symbolic Hamiltonians on several controlled synthetic and physics-inspired benchmark distributions. The paper evaluates Gaussian, many-well, XY, and $\phi^{4}$ systems, reports MSE and $R^{2}$, and shows recovered symbolic expressions, which makes the evidence fairly interpretable rather than purely black-box. The experimental setup also clearly states that the method uses normalizing flows or score models to generate likelihood/score supervision, followed by symbolic regression, with local factorization and sample perturbation as inductive biases.

However, the strongest concern is the renormalization claim. The abstract states that the method “it discovers new expressions for action
function intractable by traditional analytic approaches, thereby providing physicists with a novel tool for theoretical analysis.", and the conclusion similarly claims that the renormalization experiments discover new actions for non-perturbative regions. This is an ambitious scientific-discovery claim, but the supporting evidence appears to be mainly that the learned expressions follow expected qualitative coefficient trends across coarse-grained lattice scales. That is suggestive, but not fully convincing unless the learned effective actions are independently validated, for example, by sampling from them and comparing physical observables, correlation functions, susceptibilities, moments, or other distributional statistics against the coarse-grained data.

Another concern is that the method’s evidence depends heavily on the accuracy of the first-stage generative model. The symbolic regression stage learns from flow likelihoods or score estimates, not directly from the true unknown density. The paper does include an error-propagation discussion and shows cases where symbolic regression reduces MSE relative to the flow model, which is encouraging. But this does not fully establish that the symbolic expression recovers the true physical Hamiltonian rather than distilling the biases and errors of the learned generative model. More systematic stress tests varying flow capacity, sample size, training quality, and mode coverage would make the claim more convincing.

Also, the paper should more clearly characterize when the method succeeds or fails, rather than presenting the framework as broadly reliable. For example, the experimental results show degradation in higher-dimensional many-well cases and weaker score-based performance on the XY model.

**Requested Changes:**

- The claim that the method discovers new effective actions is currently supported mainly by qualitative coefficient trends. The authors should either soften this claim or validate the learned actions using physical observables, correlation functions, moments, or sampling-based checks against the coarse-grained data.
- Since symbolic regression learns from flow likelihoods or score estimates, errors in the generative model may propagate into the final expression. The authors should better analyze when SR recovers the true Hamiltonian versus when it distills artifacts of the learned model.
- Some settings show weaker performance, especially high-dimensional many-well examples and score-based XY results. The authors should provide guidance on when the method is expected to succeed or fail.

---

> ### Author Response · Authors · 2026-06-11
>
> 1. The claim that the method discovers new effective actions is currently supported mainly by qualitative coefficient trends. The authors should either soften this claim or validate...
>
> >We thank the reviewer for this important suggestion. To strengthen the validation, we have added quantitative physical-observable based evaluations in the revised manuscript and have also softened the corresponding claims throughout the paper.
> >
> >------
> >Specifically, for the RG $\phi^4$ experiments, we use the recovered Hamiltonians to generate new samples via HMC and compare the resulting distributions against the original coarse-grained data. We evaluate following observables, that include magnetization $M$, absolute magnetization $|M|$, and magnetic susceptibility $\chi$, for lattice sizes $32\times32$, $16\times16$, and $8\times8$.
> >
> >------
> >To quantify agreement between the coarse-grained data and samples generated from the recovered Hamiltonians, we compute (i) the Earth Mover's Distance (EMD) between the observable distributions and (ii) the percentage overlap between the corresponding histograms. The results show consistently high agreement across all observables and lattice sizes. The histogram overlap exceeds $98\%$ in every case and reaches more than $99\%$ for magnetic susceptibility at the smaller lattice sizes. Correspondingly, the EMD values remain very small, ranging from approximately $10^{-4}$ to $10^{-3}$, indicating that the generated distributions closely match the coarse-grained distributions.
> >
> >|                               | 32x32                 |                    | 16x16                 |                    | 8x8                   |                    |
> >|-------------------------------|-----------------------|--------------------|-----------------------|--------------------|-----------------------|--------------------|
> >|           Observable          | % Overlap($\uparrow$) | EMD ($\downarrow$) | % Overlap($\uparrow$) | EMD ($\downarrow$) | % Overlap($\uparrow$) | EMD ($\downarrow$) |
> >| Magnetization, M              | 98.116                | 2.28 X 10^{-4}     | 98.845                | 9.09 X 10^{-5}     | 98.759                | 8.79 X 10^{-5}     |
> >| Absolute Magnetization, \|M\| | 98.197                | 2.06 X 10^{-4}     | 98.794                | 8.69 X 10^{-5}     | 98.749                | 8.30 X 10^{-5}     |
> >| Magnetic Susceptibility       | 98.244                | 3.81 X 10^{-3}     | 99.187                | 4.47 X 10^{-4}     | 99.187                | 1.13 X 10^{-4}     |
>
>
> >|              |                |   Magnetization, M   |                    | Absolute Magnetization, \|M\| |                    | Magnetic Susceptibility |                    |
> >|:------------:|----------------|:--------------------:|:------------------:|:-----------------------------:|:------------------:|:-----------------------:|:------------------:|
> >|              |                |         Mean         | Std Dev  |              Mean             | Std Dev |           Mean          | Std Dev  |
> >| $32\times32$ | Coarsened Data | $-1.82\times10^{-6}$ |       0.0066       |             0.0053            |       0.0039       |          0.0441         |       0.0609       |
> >|              | Generated Data |  $9.09\times10^{-5}$ |       0.0068       |             0.0055            |       0.0041       |          0.0481         |       0.0681       |
> >| $16\times16$ | Coarsened Data | $-1.82\times10^{-6}$ |       0.0066       |             0.0053            |       0.0039       |          0.0110         |       0.0152       |
> >|              | Generated Data | $-3.36\times10^{-5}$ |       0.0067       |             0.0053            |       0.0040       |          0.0115         |       0.0161       |
> >|  $8\times8$  | Coarsened Data | $-1.82\times10^{-6}$ |       0.0066       |             0.0053            |       0.0039       |          0.0028         |       0.0038       |
> >|              | Generated Data | $-2.31\times10^{-5}$ |       0.0067       |             0.0053            |       0.0040       |          0.0029         |       0.0041       |
> >
> >In addition, we compare the first and second moments of these observables. For all lattice sizes considered, the means and standard deviations computed from samples generated using the recovered Hamiltonians closely match those of the coarse-grained data.
> >
> >These results provide additional evidence that the recovered Hamiltonians capture not only the symbolic structure of the coarse-grained system but also reproduce key physical observables and their distributions. Accordingly, we have revised the manuscript to present the results more cautiously. Rather than claiming that the method definitively discovers new effective actions, we now state that the recovered symbolic Hamiltonians provide interpretable approximations of the effective coarse-grained action and are validated through agreement with observable statistics and sampling-based evaluations.
> >
> >

---

> > ### Author Response · Authors · 2026-06-11
> >
> > 2. Since symbolic regression learns from flow likelihoods or score estimates, errors in the generative model may propagate into the final expression. The authors should better analyze when SR recovers the true Hamiltonian versus when it distills artifacts of the learned model.
> >
> > >We thank the reviewer for this insightful observation. We agree that the symbolic regression stage can only be as accurate as the quantities provided by the underlying generative model. Since SR operates on estimated likelihoods or score fields rather than the true Hamiltonian, errors in the generative model can indeed propagate into the recovered symbolic expression.
> > >
> > >-----
> > >In general, SR is more likely to recover the true Hamiltonian when the generative model accurately captures the target distribution and when the available samples provide sufficient coverage of the relevant state space. This is reflected in our Gaussian, $\phi^4$, and several low-dimensional benchmarks, where accurate density/score estimation leads to near-exact symbolic recovery with low coefficient error and few or no spurious terms.
> > >
> > >-----
> > >Conversely, when the generative model is inaccurate, the recovered expression may partially reflect artifacts of the learned model rather than the true underlying Hamiltonian. This situation is most evident in highly multimodal settings such as MW-64, where density estimation becomes more challenging and the recovered expressions contain larger coefficient errors and additional spurious terms. Similarly, in the score-based XY experiments, the mismatch between the circular topology of the state space and the Euclidean assumptions of standard score matching can introduce score estimation errors that subsequently affect symbolic recovery.
> > >
> > >-----
> > >To better characterize this behavior, we have added additional symbolic recovery metrics, including relative coefficient error, number of spurious terms, precision, recall, and F1 score. These metrics help distinguish successful Hamiltonian recovery from cases where the symbolic expression is influenced by artifacts introduced during the generative modeling stage. We have also expanded the discussion of these success and failure regimes in the revised manuscript.
> > >

---

> > > ### Author Response · Authors · 2026-06-11
> > >
> > > 3. Some settings show weaker performance..... The authors should provide guidance on when the method is expected to succeed or fail.
> > >
> > > >We agree that the performance of the proposed framework degrades in certain challenging regimes. To address this concern, we have expanded the discussion in the revised manuscript to clarify the conditions under which the method is expected to succeed and the factors that can limit its performance. More generally, the success of the framework depends on three key factors:
> > > >
> > > >1. **Accuracy of the generative model.**
> > > >  The first stage of the pipeline converts the unsupervised density estimation problem into a supervised learning problem using either NF or a score-based model. Consequently, the quality of the recovered expression is fundamentally limited by the accuracy of the learned density or score field. In highly multimodal distributions, NF models exhibit mode-covering behavior (Kanaujia & Arora, 2025), resulting in inaccuracies in the estimated density. These inaccuracies are subsequently inherited by the SR stage, leading to larger coefficient errors and spurious terms in the recovered expressions. This behavior is particularly evident for the highly multimodal MW-64 benchmark.
> > > >
> > > >2. **Adequate coverage of the relevant state space.**
> > > >Accurate symbolic recovery requires sufficient samples from all regions that significantly contribute to the target distribution. In practice, however, only a finite number of samples are available. For example, the MW-64 distribution contains a very large number of metastable regions ($2^{32}$ modes). Obtaining samples from all such regions is challenging, even with large training datasets. Limited coverage of low-probability or transition regions can lead to inaccuracies in both the learned generative model and the recovered symbolic expression.
> > > >
> > > >3. **Accuracy of the symbolic regression stage in systems with long-range interactions.**
> > > >The scalability of the overall framework is often determined by the SR stage rather than the generative modeling stage. As the number of relevant variables increases, the search space of candidate symbolic expressions grows rapidly, making symbolic recovery increasingly difficult. This issue is particularly important in systems with long-range interactions, where larger patches are required to capture all relevant interactions. For example, interactions extending beyond nearest neighbors may require patch sizes of $3\times3$, $4\times4$, $5\times5$, or larger. The resulting increase in the number of input variables substantially enlarges the symbolic search space and can lead to reduced recovery accuracy, larger coefficient errors, and an increased number of spurious terms.
> > > >
> > > >The weaker performance observed for the many-well benchmark can largely be attributed to the combined effects of the first two factors. The MW-64 system is both high-dimensional and strongly multimodal, with probability mass distributed across a large number of metastable regions separated by energy barriers. These characteristics make density estimation significantly more challenging and increase the amount of data required to accurately model the underlying distribution. Consequently, errors introduced during the generative modeling stage propagate to the SR stage, resulting in less accurate symbolic recovery.
> > > >
> > > >------
> > > >The score-based XY experiments present a different challenge. The underlying variables lie on a periodic manifold $(S^1)^N$ rather than in Euclidean space. Standard denoising score matching employs Gaussian perturbations and Euclidean score fields, which do not fully respect the circular topology of the state space. As a result, score estimation errors can be larger near periodic boundaries and in regions containing highly oscillatory trigonometric interactions. Since the Hamiltonian is subsequently recovered by integrating the estimated score field, these errors may accumulate and lead to reduced symbolic recovery accuracy compared with polynomial systems such as the scalar $\phi^4$ model.
> > > >
> > > >------
> > > >Based on our experiments, the framework performs best when (i) the generative model can accurately represent the target distribution, (ii) the available samples provide adequate coverage of the relevant state space, and (iii) the underlying local Hamiltonian can be represented using a moderate number of variables. Performance is expected to degrade when these conditions are violated, such as in highly multimodal systems, systems requiring very large interaction neighborhoods, or systems whose geometry is not naturally aligned with the assumptions of the underlying generative model.
> > > >
> > > >------
> > > >Importantly, even in these challenging settings, the recovered expressions often remain useful as compact and interpretable approximations of the underlying energy landscape. However, they should be viewed as approximate effective models rather than exact symbolic recoveries of the true Hamiltonian.